# Neural representations of naturalistic events are updated as our understanding of the past changes

Asieh Zadbood[1]*, Samuel Nastase[2], Janice Chen[3], Kenneth A Norman[2], Uri Hasson[2]

[1]Department of Psychology, Columbia University, New York, United States; [2]Princeton Neuroscience Institute and Department of Psychology, Princeton University, Princeton, United States; [3]Department of Psychological and Brain Sciences, Johns Hopkins University, Baltimore, United States

**Abstract** The brain actively reshapes our understanding of past events in light of new incoming information. In the current study, we ask how the brain supports this updating process during the encoding and recall of naturalistic stimuli. One group of participants watched a movie ('The Sixth Sense') with a cinematic 'twist' at the end that dramatically changed the interpretation of previous events. Next, participants were asked to verbally recall the movie events, taking into account the new 'twist' information. Most participants updated their recall to incorporate the twist. Two additional groups recalled the movie without having to update their memories during recall: one group never saw the twist; another group was exposed to the twist prior to the beginning of the movie, and thus the twist information was incorporated both during encoding and recall. We found that providing participants with information about the twist beforehand altered neural response patterns during movie-viewing in the default mode network (DMN). Moreover, presenting participants with the twist at the end of the movie changed the neural representation of the previously-encoded information during recall in a subset of DMN regions. Further evidence for this transformation was obtained by comparing the neural activation patterns during encoding and recall and correlating them with behavioral signatures of memory updating. Our results demonstrate that neural representations of past events encoded in the DMN are dynamically integrated with new information that reshapes our understanding in natural contexts.

*For correspondence:
az2604@columbia.edu

## Editor's evaluation

This study presents an important extension of recent work investigating the encoding and recall of narratives in the default mode network. The design is clever, allowing the authors to conduct multiple targeted analyses. The results are compelling and will be of interest to cognitive neuroscientists working on memory and naturalistic paradigms.

## Introduction

In a constantly changing world, it is critical to update prior beliefs and memories in light of new circumstances. As new information arrives, we often need to update previously encoded information in the brain retrospectively. Imagine discovering that a longtime friend has lied to you about something important. You might automatically start looking back and reinterpreting their behavior, perhaps finding different motives for their past actions. This updated understanding of the past will assist you in your future interactions with that friend. Importantly, updating representations of real-world events does not necessarily involve rewriting

or erasing the content of the previous memory for the event – it can also include adding new information that alters one's overall interpretation of what happened. In this paper, we use the term 'memory updating' to refer to this process of updating representations of past events based on new information. To effectively support 'memory updating', the episodic memory system must be capable of modifying stored representations in light of new incoming information. Under this framework, memories are dynamic entities that can be reorganized or reconstructed even after encoding takes place (*Bartlett and Burt, 1933*; *Conway and Pleydell-Pearce, 2000*; *Hassabis and Maguire, 2007*; *Schacter et al., 1998*; *Schacter, 2012*).

Research in the last few decades suggests that memories are malleable to modification when they are reactivated (*Przybyslawski and Sara, 1997*), and relevant new information is presented (*Besnard et al., 2012*; *Hupbach et al., 2015*; *Nader and Einarsson, 2010*; *Sinclair and Barense, 2019*). Behavioral paradigms using a retroactive interference design have been widely used to study post-encoding changes in human memory (e.g. *Lee et al., 2017*; *Hupbach et al., 2015*; *Samide and Ritchey, 2020*; *Scully et al., 2017*). Only a subset of studies, however, have investigated changes in the *content* of memory, as opposed to the weakening or strengthening of old memories (*Dongaonkar et al., 2013*; *Hupbach et al., 2007*). At the neural level, changes in the functional connectivity of mPFC and amygdala circuitry have been associated with post-retrieval fear extinction (*Feng et al., 2016*; *Schiller et al., 2013*). These experimental studies have clinical significance and provide valuable insight into the behavioral and neural substrates of memory updating in humans. However, it is unclear how findings obtained using tightly controlled paradigms and isolated stimuli generalize to memory updating in everyday life (*Nastase et al., 2020*). In the present work, we introduce a naturalistic interference-based design that resembles our real-world experiences where new information obtained post-encoding is not compatible with previously encoded events. Using an audiovisual movie and verbal recall, we aim to utilize recent advances in naturalistic neuroimaging to study how memories are reshaped to incorporate new incoming information.

The brain's default mode network (DMN)—comprising the posterior medial cortex, medial prefrontal cortex, temporoparietal junction, and parts of anterior temporal cortex—was originally described as an intrinsic or 'task-negative' network, activated when participants are not engaged with external stimuli (*Raichle et al., 2001*; *Buckner et al., 2008*). This observation led to a large body of work showing that the DMN is an important hub for supporting internally driven tasks such as memory retrieval, imagination, future planning, theory of mind, and creating and updating situation models (*Svoboda et al., 2006*; *Addis et al., 2007*; *Hassabis and Maguire, 2007*; *Hassabis and Maguire, 2009*; *Schacter and Addis, 2007*; *Szpunar et al., 2007*; *Spreng et al., 2009*; *Koster-Hale and Saxe, 2013*; *Ranganath and Ritchey, 2012*). However, it is not fully understood how this network contributes to these varying functions, and in particular—the focus of the present study—memory processes. Activation of this network during 'offline' periods has been proposed to play a role in the consolidation of memories through replay (*Kaefer et al., 2022*). Interestingly, prior work has also shown that the DMN is reliably engaged during 'online' processing (encoding) of continuous rich dynamic stimuli such as movies and audio stories (*Stephens et al., 2013*; *Hasson et al., 2008*). Regions in this network have been shown to have long 'temporal receptive windows' (*Hasson et al., 2008*; *Lerner et al., 2011*; *Chang et al., 2022*), meaning that they integrate and retain high-level information that accumulates over the course of extended timescales (e.g. scenes in movies, paragraphs in text) to support comprehension. This combination of processing characteristics suggests that the DMN integrates past and new knowledge, as regions in this network have access to incoming sensory input, recent active memories, and remote long-term memories or semantic knowledge (*Yeshurun et al., 2021*; *Hasson et al., 2015*). These integration processes feature in many of the 'constructive' processes attributed to DMN such as imagination, future planning, mentalizing, and updating situation models (*Schacter and Addis, 2007*; *Ranganath and Ritchey, 2012*). Notably, constructive processes are highly relevant to real-world memory updating, which involves selecting and combining the relevant parts of old and new memories. Recent work has shown that neural patterns during encoding and recall of naturalistic stimuli (movies) are reliably similar across participants in this network (*Chen et al., 2017*; *Oedekoven et al., 2017*; *Zadbood et al., 2017*; see *Bird, 2020* for a review of recent naturalistic studies on memory), and the DMN displays distinct neural activity when listening to the same story with different perspectives (*Yeshurun et al., 2017*). Building on this foundation of prior work on the DMN, we asked whether we could find neural evidence for the retroactive influence of new knowledge on past memories.

In the current work, using a novel naturalistic paradigm intended to simulate a real-life situation of adaptive memory updating, we asked how new information changes the neural representations in the DMN during the recall of prior events. To answer this question, we used a popular Hollywood-style film titled 'The

Sixth Sense' (M. Night Shyamalan, 1999), which contains a dramatic twist in the final scene. [Spoiler alert!] The movie depicts the story of a clinical psychologist treating a child who claims to see ghosts. In the final scene, it is revealed that the doctor was in fact, a ghost himself throughout the movie. Therefore, there are two coherent interpretations of the movie: the *Doctor* (or *naive*) interpretation (labeled D in *Figure 1*), which is typically held by viewers up until they encounter the 'twist ending'; and the *Ghost* (or *spoiled*) interpretation (labeled G in *Figure 1*), which is held by viewers after they learn about the twist. In this setting, memory updating is operationalized as the transition from the *Doctor* (D) interpretation to the *Ghost* (G) interpretation.

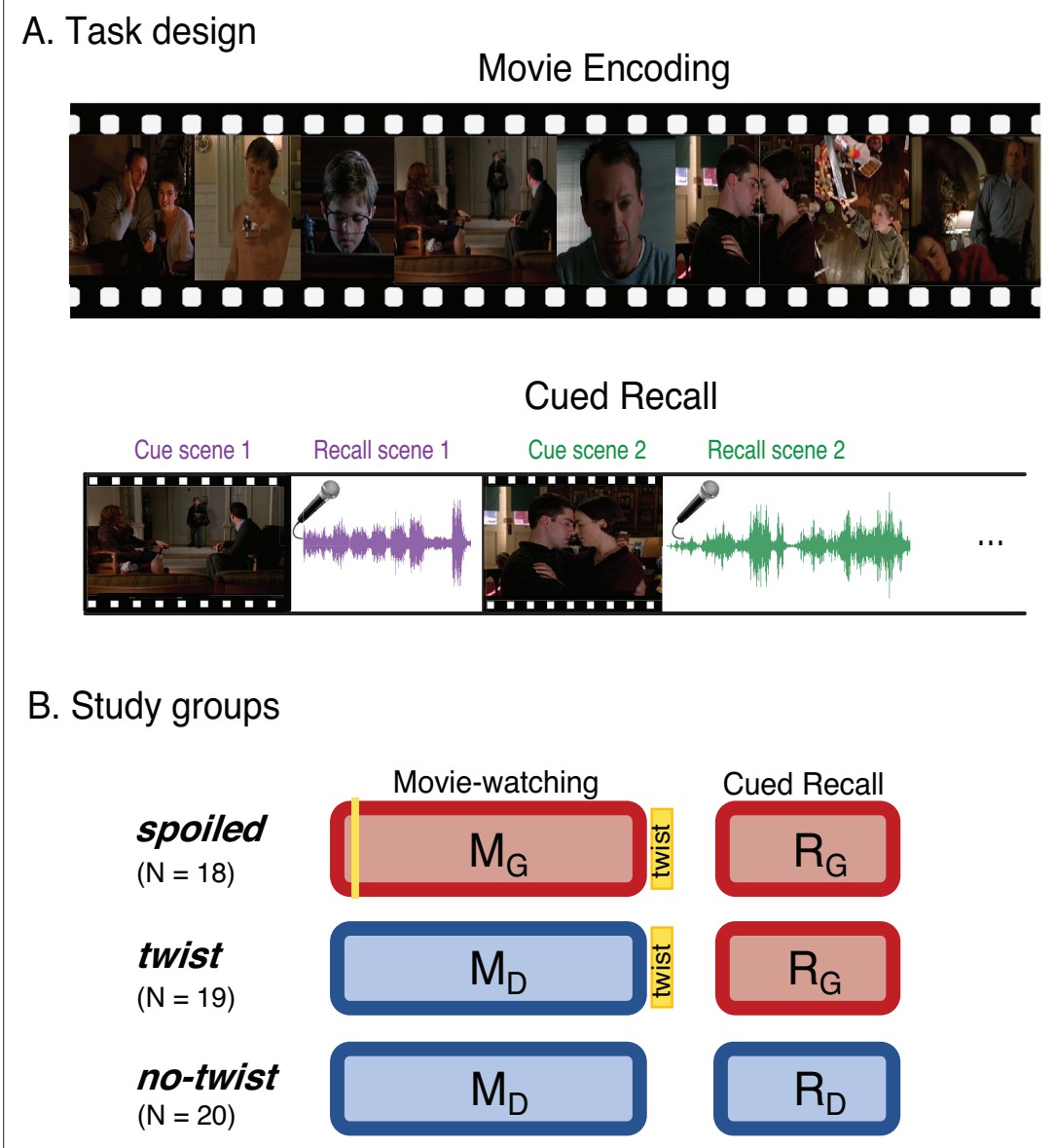

**Figure 1.** Experimental design. (**A**) Participants watched edited versions of the movie and performed a scene-by-scene cued verbal recall task in the scanner. (**B**) Experimental groups. Red boxes refer to the *Ghost* interpretation, and blue boxes refer to the *Doctor* interpretation. The 'twist' group (middle row) is the main experimental group that encodes the movie with *Doctor* interpretation (left blue box) but recalls it with *Ghost* interpretation (right red box)—essentially following the narrative as intended by the filmmaker. The two additional groups keep the same interpretation across the encoding and recall: the 'spoiled' group receives a spoiler at the beginning, thus encoding the movie and performing the recall task with the red *Ghost* interpretation, whereas the 'no-twist' group never receives the twist and therefore encodes the movie and performs the recall task under the blue *Doctor* interpretation.

Our study design hinges on the hypothesis that participants who received the twist and are aware that the doctor is a ghost might have distinct neural representations of the events from those who encoded the movie while ignorant of the twist. Importantly, we predicted that encountering the twist after encoding the movie would initiate a retrospective update in the interpretation of the encoded movie and that this update would be reflected in both verbal recall and patterns of brain activity during remembering. In contrast, the neural representations of the events in the movie will remain unchanged during recall in subjects who do not need to update their memories during recall (i.e. in subjects in the 'no-twist' condition who are only aware of the D interpretation, or subjects in the 'spoiler' condition who knew all along about the G interpretation).

In a large set of regions in the DMN, we found that context changed how the movie was encoded into memory. In other words, the neural representations for each event in the movie were different for viewers who believed the doctor was alive versus viewers who believed the doctor was a ghost. Furthermore, in several DMN regions, we found that neural representations were updated during recall for viewers who learned that the doctor was a ghost after watching the movie. Together these results suggest that areas in the default mode network are actively updating the neural representations as they integrate incoming information with prior knowledge.

## Results

Three distinct experimental groups watched concise versions of a popular Hollywood-style film titled "The Sixth Sense" (M. Night Shyamalan, 1999) in the fMRI scanner (*Figure 1B*, left column). Following the movie viewing all three groups were asked to freely recall the movie in the scanner (*Figure 1B*, left column). Participants in the main group (the 'twist' condition, *Figure 1B*, middle row), watched the movie with the twist scene *at the end*. Therefore, they watched the movie naive to the true nature of the doctor (*Movie-Doctor* or $M_D$). During their recall, however, they were aware of the twist information and could use it to update their memory (*Recall-Ghost* or $R_G$). In order to identify interpretation-specific neural patterns, we needed two comparison conditions: the *Movie-Ghost* ($M_G$) condition during viewing, and the *Recall-Doctor* ($R_D$) condition during recall. Therefore, we introduced two other groups to the study: participants in one group (the 'spoiled' condition; *Figure 1B*, top row) were exposed to the twist at the beginning of the movie. This group watched and recalled the movie knowing that the doctor was a ghost ($M_G$ and $R_G$). The other group (the 'no-twist' condition; *Figure 1B*, bottom row) never received the twist information throughout encoding and remained naive to the true nature of the doctor in both their encoding and recall ($M_D$ and $R_D$). This design allowed us to compare the behavioral and neural patterns of response in participants across the two interpretations. We compared the patterns of neural responses in the 'twist' group with the patterns in the 'spoiled' and 'no-twist' groups during encoding and recall. We predicted that the 'twist' group would be more similar to the 'no-twist' group during encoding (both having the *Doctor* interpretation) but more similar to the 'spoiled' group during recall (both having the *Ghost* interpretation). Moreover, we asked whether the memory updating would make the recall of the 'twist' group more similar to the encoding of the 'spoiled' group (see the 'prediction legends' in Figures 2 and 3).

We used intersubject pattern similarity analysis (intersubject pattern correlation: pISC, see Methods) to compare the neural event representations between groups. This analytic approach is motivated by prior work showing that slowly-evolving activity patterns in DMN represent event-level information (*Baldassano et al., 2018*) and that pISC captures scene-specific pattern similarity between groups who watched the same movie, groups who verbally recalled the same story (in their own words), and across viewing and recall of the same scenes (*Chen et al., 2017*; *Zadbood et al., 2017*). In this analysis, scene-specific neural patterns during encoding of the movie were obtained by averaging data across time within each scene in each subject (*Chen et al., 2017*; *Zadbood et al., 2017*). For the cued recall data, we ran a GLM analysis (*Mumford et al., 2012*) to capture responses corresponding to the recall of single events. fMRI responses averaged across time points within an event (or estimated from the GLM) for each ROI served as the *spatial response patterns* (i.e. neural event representations) that were then compared across groups using pISC. We then compared pairs of pattern similarity correlations based on our hypotheses. For example, we hypothesized that the updated recall in the 'twist' group would be more similar to the recall of the 'spoiled' group (as both groups have the *Ghost* interpretation during the recall) than of the 'no-twist' group (who have the *Doctor* interpretation). To test this hypothesis, in each ROI, we measured pISC once between 'twist' and

'spoiled' groups and once between 'twist' and 'no-twist' groups. We then compared these two sets of pattern similarity values to quantify which two groups' neural event representations were more similar. We focused our analyses on a predetermined selection of movie scenes (i.e. 7 'critical scenes' out of 18 total scenes) in which the *Doctor* or *Ghost* interpretation of the main character in the movie would dramatically change the overall interpretation of those scenes. Selection of these scenes was based on ratings from five raters asked to quantify the influence of the twist on the interpretation of each scene (see Methods).

After watching the movie, participants performed a cued-recall task in which they watched a few seconds of the beginning of all movie scenes (18 scenes) and were asked to describe what happened next in that scene. The recall task was identical across the three experimental conditions. Participants were highly accurate in recognizing the corresponding scenes from the movie cues (94% accuracy in the 'twist' group, 93% in the 'spoiled' group, and 97% in the 'no-twist'group). Only the scenes that were correctly recalled were included in the neural analyses. The content of recall was evaluated using two separate measures assigned by human raters. *Memory score* assessed the quality and detail of memory. *Twist score* assessed whether the twist information was incorporated into the recall and ranged from 1 (the recall purely reflected the *Doctor* interpretation) to 5 (the recall purely reflected the *Ghost* interpretation). Memory score and twist score were expected to capture different aspects of the recall behavior; for example a detailed recall of the original scene about the doctor treating the child (high memory score) may not include information about the doctor being a ghost (low twist score). Indeed, there was no significant correlation between memory scores and twist scores across participants ($r$=0.07, $p$=0.56). If participants were unaware of the twist or did not incorporate it into their recall at all, we would expect the average twist score of the critical scenes to be approximately equal to 1 ('purely reflects the *Doctor* interpretation'). In the main experimental group ('twist' group), 14 out of 19 participants scored above 2 (median score = 3.25) on the twist score, indicating that they incorporated the new interpretation into their recall. Importantly, the 'twist' group (twist score: M=3.16, SD = 1.03) exhibited a significantly higher twist score ($t(37)$ = 6.37, $p$<0.001) than the 'no-twist' group (twist score: M=1.65, SD = 0.22). Note that these two groups had no knowledge of the twist when they encoded the movie. Therefore, this result confirms that participants in the 'twist group' updated their memories of the movie to incorporate the twist. No significant difference ($t(35)$ = 1.46, $p$=0.15) was observed between the twist score of the 'twist' group and the 'spoiled' group (twist score: M=2.72, SD = 0.74). This finding suggests that the 'twist' group recalled the movie more similarly to the group that knew the twist while watching the movie.

## Memory update in recall behavior

A surprising observation during the analysis of the behavioral recall in the 'twist' condition was that most participants talked about *both* interpretations of the movie scenes in many of the recalled scenes (this pattern was observed in the recall of the 'spoiled' group as well). Thus, it appeared that participants kept both interpretations in mind during the recall, instead of overwriting the *Doctor* representation with the *Ghost* representation. These recalls were typically structured as, "Initially I thought that... but now I know that...". Interestingly, some instances of this recall behavior were also observed in the 'spoiled' group, who had watched the movie knowing the doctor is a ghost (e.g. "You could think that... but I knew that..."). This suggests that the neural representations supporting recall in the 'twist' and 'spoiled' groups included *both* the original (*Doctor*) and updated (*Ghost*) interpretations, which could make differentiating these representations in the neural analysis more challenging (see Discussion).

## Neural representation of the twist information during movie-viewing

First, we set out to test how contextual knowledge about the twist modifies the neural patterns in the DMN during the encoding of the movie into memory. As encoding conditions are directly compared, we refer to this analysis as encoding-encoding throughout the paper. We compared the spatially distributed neural activity patterns elicited during movie-viewing (encoding) in the 'twist' group ($M_D$) to the activity patterns obtained during encoding in the 'no-twist' group ($M_D$) and the 'spoiled' group ($M_G$). We hypothesized that within the regions of the brain that are sensitive to different interpretations, the pattern similarity between the 'twist' group ($M_D$) and the 'no-twist' group ($M_D$) should be higher than the similarity between the 'twist' group ($M_D$) and the 'spoiled' group ($M_G$) (*Figure 2A*, prediction legends).

Indeed, there was significantly greater intersubject pattern correlation in parts of the DMN between the 'twist' and 'no-twist' experimental groups (who had a similar $M_D$ interpretation of the movie during encoding) than across experimental groups with opposing interpretations ($M_D$ versus $M_G$). These areas included the

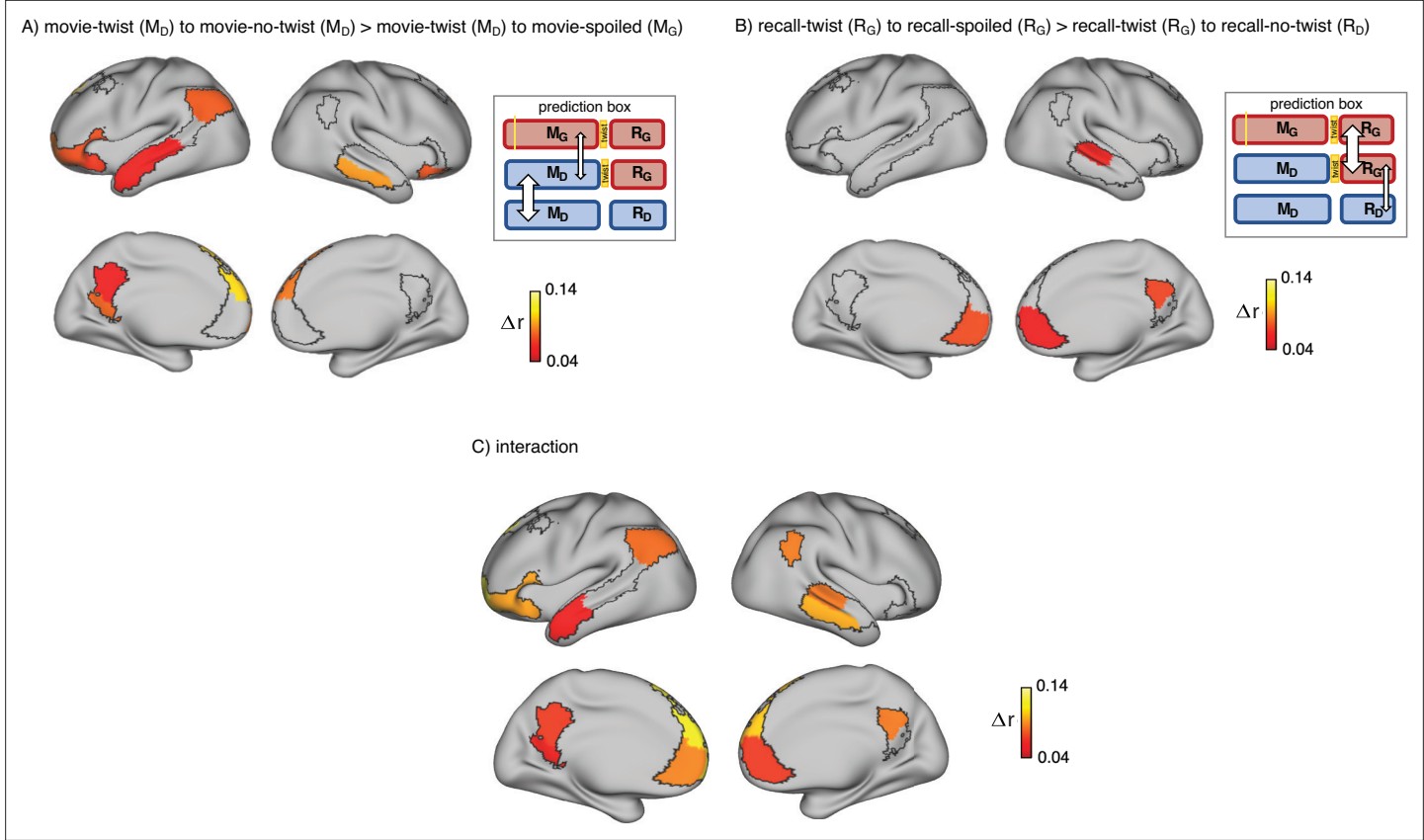

**Figure 2.** Brain regions coding for story interpretation at encoding and recall. 'Prediction legends' depict the predicted pattern of correlations between groups based on our hypotheses. (**A**) Areas with significantly greater intersubject pattern correlation between groups who encoded the movie with the same interpretation (*Doctor*). (**B**) Areas with significantly greater intersubject pattern correlation between groups who recalled the movie with the same interpretation (*Ghost*). (**C**) Areas with a significant interaction effect, indicating a change in interpretation between encoding and recall (see 'Intersubject pattern correlation (pISC) analysis' in Methods). Statistical significance was assessed using a nonparametric randomization test, FDR corrected p<0.05.

dorsal and lateral PFC, left precuneus, left retrosplenial cortex, left angular gyrus, middle temporal cortex, left superior temporal cortex, and left temporal pole (*Figure 2A*). These results fit with previous findings demonstrating that the timecourse of brain responses in DMN regions reflects different perspectives when listening to a spoken narrative (*Yeshurun et al., 2017*). Our results extend these findings by showing that different interpretations are discriminable in *spatial* response patterns measured while viewing audiovisual movie stimuli.

## Neural representation of the twist information during cued recall

Results from the encoding phase suggest that regions in DMN exhibit different patterns of neural response to *Ghost* vs. *Doctor* interpretations. In the next step, we sought to measure memory updating, which we define as a shift during recall from the neural patterns associated with the *Doctor* interpretation to incorporate information associated with *Ghost* interpretation. Since recall conditions are directly compared, we refer to this analysis as recall-recall. As described earlier, the analysis of recall behavior suggests that participants in the 'twist' condition utilized the twist information to update their recall of the movie. Hence, we ask whether the neural patterns observed during recall would reflect these changes. We predicted that the 'no-twist' group and the 'spoiled' group would keep the same interpretation of the movie during encoding and recall ($M_D$ to $R_D$ in the 'no-twist' group and $M_G$ to $R_G$ in the 'spoiled' group). However, in the 'twist' group, we expected to observe an update during recall to accommodate the twist information ($M_D$ to $R_G$). Therefore, we hypothesized that, during recall, the neural patterns for the 'twist' group might shift from being more similar to the 'no-twist' group as observed during encoding to be more similar to the neural patterns in the 'spoiled' group during recall (*Figure 2B* – prediction legends).

Indeed, as subjects recalled the movie in the scanner, there was significantly greater intersubject pattern correlation in parts of the DMN between the 'twist' and 'spoiled' experimental groups (who believed that the doctor is a ghost: $R_G$) than across the 'twist' and 'no-twist' groups (who had opposing interpretations: $R_G$ versus $R_D$). These areas included the ventromedial prefrontal cortex (vmPFC), right precuneus, and right superior temporal cortex (*Figure 2B*). In addition, we ran an interaction analysis to further emphasize the reversal of neural similarity during encoding and recall (see Methods). This analysis highlights a large set of DMN regions, including medial, dorsal, and lateral PFC, precuneus, left retrosplenial cortex, angular gyrus, right superior and middle temporal cortex, and left temporal pole, where neural patterns in the 'twist' group were relatively more similar to the *Ghost* (vs. *Doctor*) interpretation at recall than at encoding (*Figure 2C*).

To test whether our reported results were mainly driven by the similarities and differences in multivariate spatial patterns of neural representations, as opposed to by univariate regional-average response magnitudes, we ran a univariate analysis in each ROI. This analysis revealed no significant effect of group ('spoiled', 'twist', 'no-twist') or interaction between group and condition (movie, recall) (*Appendix 1—table 1*, see Methods for details).

## Relationship between the neural representations during encoding and recall

To provide further neural evidence for the shift from *Doctor* interpretation during encoding to *Ghost* interpretation during recall in the 'twist' group, we directly compared the brain responses elicited during encoding and recall (encoding-recall analyses). *Chen et al., 2017* have demonstrated that, across free recall of a movie, neural patterns are reinstated in DMN. In addition, these scene-specific neural patterns changed between encoding and recall in a systematic manner across individuals (*Chen et al., 2017*). We hypothesized that updating one's interpretation to incorporate twist information might alter the neural representations during recall, such that they become more similar to the neural patterns elicited during encoding of the spoiled movie.

We tested this hypothesis in two ways. First, we predicted that (*Figure 3A*, prediction legend) the neural pattern similarity between recall in the 'twist' group and encoding in the 'spoiled' group ($R_G$ to $M_G$) would be higher than the pattern similarity between recall in the 'no-twist' group and encoding of the 'spoiled' group ($R_D$ to $M_G$). Our analysis confirmed this prediction in the left angular gyrus, left dorsomedial PFC, and right middle temporal cortex (*Figure 3A*).

Second, if participants in the 'twist' group were to *fully* update their interpretation at recall from *Doctor* to *Ghost*, we would expect activity patterns during recall in the 'twist' group to be more similar to encoding in the 'spoiled' group ($R_G$ to $M_G$) compared to encoding in their own ('twist') group ($R_G$ to $M_D$) (*Figure 3B*, prediction legends). When we looked for regions showing this effect, we found weak effects in the predicted direction in the left angular gyrus, left frontal pole, and right anterior temporal ROIs (note that all of these comparisons were performed across participants; see Methods for details); however, these effects did not

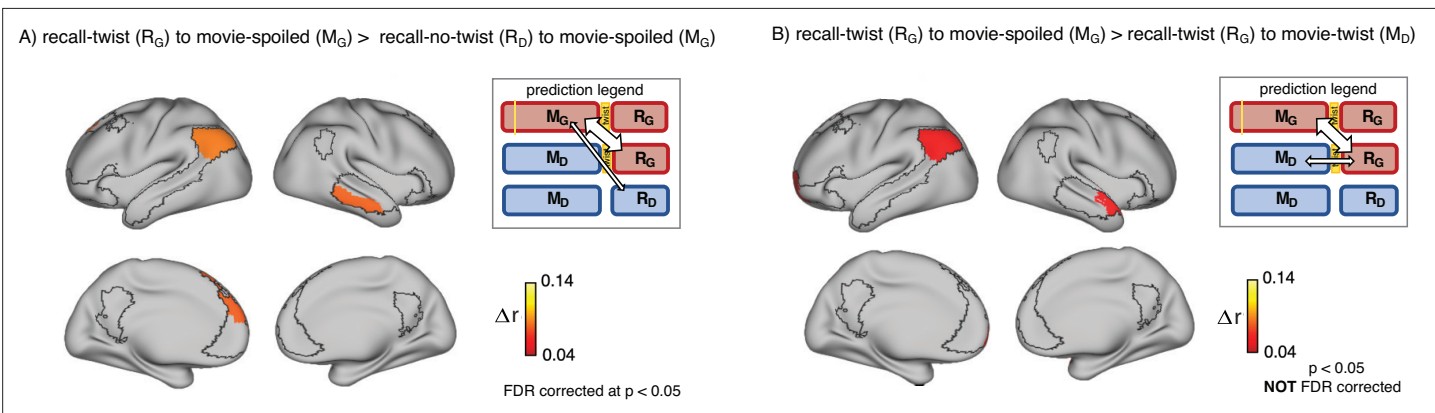

A) recall-twist ($R_G$) to movie-spoiled ($M_G$) > recall-no-twist ($R_D$) to movie-spoiled ($M_G$)

B) recall-twist ($R_G$) to movie-spoiled ($M_G$) > recall-twist ($R_G$) to movie-twist ($M_D$)

prediction legend

FDR corrected at p < 0.05

p < 0.05
**NOT** FDR corrected

**Figure 3.** Encoding-retrieval similarity analyses to test our memory updating predictions. "Prediction legends" depict the predicted pattern of correlations between groups based on our hypotheses. (**A**) Areas where intersubject pattern correlations were significantly greater when comparing updated recall ($R_G$) to spoiled encoding ($M_G$) than when comparing naive recalls ($R_D$) to spoiled encoding ($M_G$). (**B**) Areas where intersubject pattern correlations between updated recall ($R_G$) and spoiled encoding ($M_G$) were greater than between updated recall ($R_G$) and naive encoding ($M_D$); note that these results were not significant after correction for multiple tests.

survive correction for multiple comparisons at an FDR-corrected p<0.05 (*Figure 3B*). The most straightforward interpretation of these weak effects is, in general, 'twist' group participants did *not* fully update their interpretations; that is, there may have been some lingering memory of the *Doctor* interpretation in the 'twist' group in some participants even after they were exposed to *Ghost* interpretation and updated their memory.

To test this hypothesis, we ran an exploratory analysis where we correlated neural pattern change (i.e. the degree to which the neural pattern at recall matched the *Doctor* or *Ghost* encoding pattern) with behavioral twist scores (i.e. how much each subject discussed the twist during recall) across participants in the 'twist' group, in each DMN ROI (*Appendix 1—figure 1*). If weak neural pattern change effects are due to incomplete memory updating, we would expect to see a positive correlation between these measures. We observed a positive correlation between the neural and behavioral indices of memory update in posterior regions of the DMN, including precuneus and angular gyrus. The right precuneus ROI exhibited a notable relationship (*r*=0.62); however, this did not survive FDR correction across ROIs.

## The role of scene content

In the prior analyses, we focused on 'critical scenes', selected based on ratings from five raters who quantified the influence of the twist on the interpretation of each scene (see Methods). An independent post-experiment analysis of the verbal recall behavior of the fMRI participants yielded 'twist scores' that were also highest for these scenes; that is, the expected and perceived effect of twist information on recall behavior were found to match. In our next analysis, we asked whether the neural event representations reflect these differences in the twist-related content of the scenes. In other words, are the 'critical scenes' with highly twist-dependent interpretations truly *critical* for our observed effects?

To answer this question, we re-ran our main encoding-encoding and recall-recall pISC analysis in each DMN ROI (*Figures 2–3*). We calculated interaction indices (*Figure 2C*) first by including all scenes, and second by including only the 11 non-critical scenes. To better compare the effect of including different subsets of scenes to our original results, in *Figure 4* we show the results in 15 ROIs that exhibited meaningful effects in our main analyses (*Figure 2C*). *Figure 4A* demonstrates that 'critical scenes' yielded higher interaction indices compared to all scenes or non-critical scenes across all ROIs. The interaction score across all DMN ROIs was significantly higher in 'critical scenes' than all scenes (t(23) = 7.19, p=2.53 x 10⁻⁷) and non-critical scenes (t(23) = 7.3, p=1.95 x 10⁻⁷). These results show that critical scenes are indeed responsible for the observed pISC differences across groups.

Next, to determine whether scene-specific neural event representations—as opposed to coarser differences in general mental state across all scenes with similar interpretations—drive our observed pISC differences, we shuffled the labels of critical scenes within each group before calculating and comparing pISC across groups. By repeating this procedure 1000 times and recalculating the interaction index at each iteration, we constructed a null distribution of interaction indices for shuffled critical scenes (light magenta distributions in *Figure 4B*). In 12 out of 24 DMN regions, interaction indices were statistically significant based on the shuffled-scene distribution (p<0.025, FDR controlled at q<.05). All of these 12 regions were among the ROIs that showed meaningful effects in our original analysis (*Figure 2C*). Regions with significant scene-specific interaction effects are marked as blue dots with black borders in *Figure 4B*. Overall, the findings from this analysis confirm that our results are driven by changes to scene-specific representations.

To further evaluate the relationship between scene-specific twist information in the brain and behavior, we ran an exploratory analysis which was focused on the changes in the neural event representations during recall of the 'twist' group and their corresponding recall behavior. We ran the same pISC procedure described in *Figures 2B and 3B*. However, we did not average the pISC differences across scenes. In the recall-recall analysis, this procedure yielded 18 values (one for each scene, averaged across participants) indicating whether the neural event representations during updated recall were more similar to the spoiled recall or the naïve recall for a given scene. We then correlated these values with the twist score data (based on ratings of verbal recall) for each scene averaged across participants (*Appendix 1—figure 4A*). None of these correlations were significant after correction for multiple tests; however, the four ROIs with significant effects in the main recall-recall analysis (*Figure 2B*) all showed positive correlations, particularly left mPFC. We repeated the same procedure to compute the relationship between scene-level neural event representations and behavioral twist scores in the encoding-recall analysis (*Figure 3B*). In the original analysis (*Figure 3B*), we

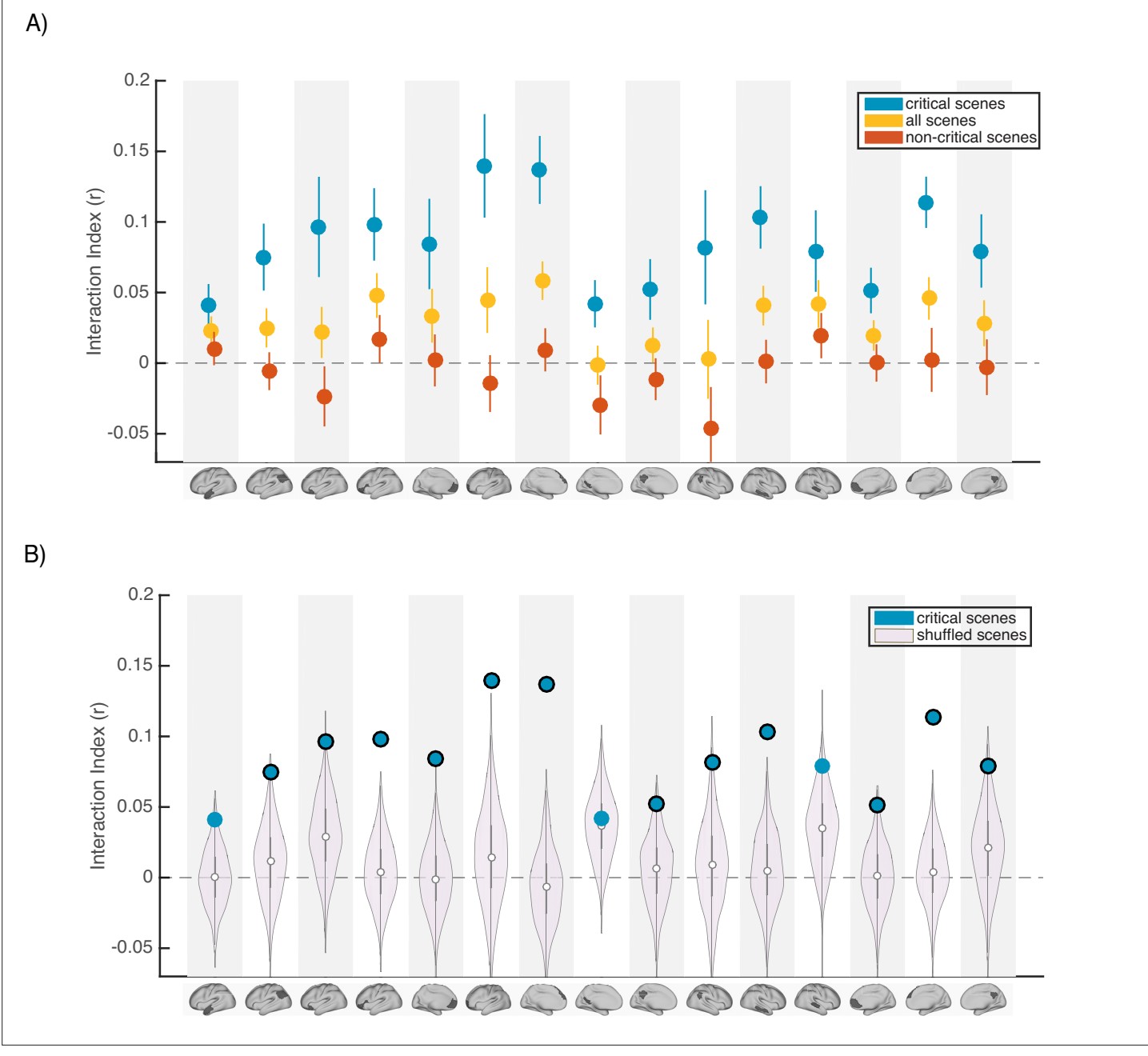

**Figure 4.** Scene content analyses. Interaction indices for 15 regions with meaningful results in the original interaction analysis (*Figure 2C*). Note that the analyses were performed in all 24 DMN regions and this set is selected for visualization, to facilitate comparison with the prior analysis. (**A**) Interaction indices were computed separately for 'critical scenes' (blue dots), 'non-critical scenes' (red dots), or all scenes (yellow dots). Error bars depict standard error of mean (N = 19). (**B**) Light magenta distributions depict interaction values calculated after shuffling critical scene labels 1000 times. Blue dots indicate the actual (non-shuffled) interaction index; blue dots marked with black borders are statistically significant based on the null distribution (FDR controlled at q<0.05).

evaluated whether the neural representations of updated recall in the 'twist' group were more similar to encoding in the 'spoiled' group than their own naïve encoding. Here, we followed the same procedure of correlating scene-level outputs with scene-level twist scores (*Appendix 1—figure 4B*). Again, none of these correlations were significant after correcting for multiple tests, but the three regions with (uncorrected) effects in the original analysis (*Figure 3B*) all displayed positive correlations with twist score. As we did not find strong results in this exploratory set of analyses, we refrain from over-interpreting them. However, they appear to match the direction of our main analyses; with greater

statistical power analyses of this sort may provide insights into how neural event representations are updated in a scene-specific manner.

## Changes in neural representations beyond the DMN

We focused our core analyses on regions of the default mode network. Prior work has shown that multimodal neural representations of naturalistic events (e.g. movie scenes) are similar across encoding (movie-watching or story-listening) and verbal recall of the same events in the DMN (*Chen et al., 2017*; *Zadbood et al., 2017*). Therefore, in the current work we hypothesized that retrospective changes in the neural representations of events as the narrative interpretation shifts would be observed in the DMN. We did not, for example, expect to observe such effects in lower-level sensory regions, where neural activity differs dramatically for movie-viewing and verbal recall. To be thorough, we ran the same set of analyses we performed in the DMN (*Figures 2–3*) in regions of the visual and somatomotor networks extracted from the same atlas parcellation (*Schaefer et al., 2018*). Our results revealed larger overall differences in DMN than in visual and somatosensory networks for the key comparisons discussed previously (*Appendix 1—figure 2*). In particular, the only regions showing significant differences in pISC in recall-recall and encoding-recall comparisons (p<0.01, uncorrected) were located in the DMN. We did not observe a notable difference between DMN and the two other networks when comparing recall 'twist' to movie 'spoiled' and recall 'twist' to movie 'twist' ($R_G - M_G > R_G - M_D$) which is consistent with the weak effect in the original comparison (*Figure 3B*). In the encoding-encoding comparison, several ROIs from the visual and somatomotor networks showed relatively strong effects as well (see Discussion).

In addition, we qualitatively reproduced our results by performing an ROI-based whole brain analysis (*Appendix 1—figure 3*, p<0.01 uncorrected). This analysis confirmed the importance of DMN regions for updating neural event representations. However, strong differences in pISC in the hypothesized direction were also observed in a handful of other non-DMN regions, including ROIs partly overlapping with anterior cingulate cortex and dorsolateral prefrontal cortex (see Discussion).

## Discussion

Using a novel naturalistic paradigm that prompted participants to update their previously-encoded memories, we studied how new information can retrospectively change the event representations in the default mode network. At encoding, a widespread network of frontal, parietal, and temporal regions exhibited significantly higher pattern similarity between groups in which participants had the same interpretation of the movie (naive to the twist; see *Figure 2A*). This result demonstrates how a belief about the identity of the doctor (which can broadly be construed as the context or the state of mind of the observer) can shape the encoding processes of new information (the same movie) into memory. But information is not only shaped by context during encoding, as stored representations must also be amenable to change as the context changes at a later stage. Indeed, our unique paradigm allows us to see how the patterns of stored representations change, as we learn about the twist in the movie. In particular, the neural patterns during recall changed in the twist condition to better match the neural patterns in the spoiled condition observed during recall in the ventromedial PFC, right precuneus, and temporal cortex (see *Figure 2B*). Furthermore, numerous areas throughout the DMN showed a significant interaction whereby neural patterns in the 'twist' group became relatively more similar to patterns from the 'spoiled' *Ghost* group (compared to the 'no-twist' *Doctor* group) at recall (compared to encoding; *Figure 2C*).

We also found evidence for memory updating by directly comparing patterns from encoding and retrieval. In the left angular gyrus, left dorsomedial PFC, and right middle temporal cortex, viewing the twist at the end of the movie (vs. not viewing the twist) resulted in neural patterns at recall becoming more similar to the 'spoiled' *Ghost* encoding patterns (*Figure 3A*). In some regions, this updating effect led to 'twist' recall patterns being numerically more similar to the 'spoiled' encoding patterns than to encoding patterns from the 'twist' condition, but this effect did not survive multiple comparisons correction (*Figure 3B*). We suggested that the weakness of this effect may be attributable to some participants not fully discarding the *Doctor* interpretation when they update their interpretation; in line with this, an exploratory analysis showed that—in some DMN ROIs—the degree of neural change was nominally correlated (across participants) with behavioral 'twist scores' capturing how

strongly a participant's recall was influenced by the twist (*Appendix 1—figure 1*; these exploratory correlations did not survive multiple comparisons correction). Taken together, our results provide further evidence for the involvement of DMN regions in integrating new information with prior knowledge to form distinct high-level event representations. In particular, we suggest a subset of core DMN regions are implicated in representing changes in event interpretations during memory updating.

The default mode network, traditionally known to support internally oriented processes, is now considered a major hub for actively processing incoming external information and integrating it with prior knowledge in the social world (*Yeshurun et al., 2021*). Our experimental design targets naturalistic event representations unfolding over seconds to minutes. There have been many studies to date corroborating the discovery of a cortical hierarchy of increasing temporal receptive windows where high-level event representations are encoded at the top of the hierarchy—in the DMN (*Hasson et al., 2008*; *Lerner et al., 2011*; *Hasson et al., 2015*; *Baldassano et al., 2018*; etc). This network is involved in episodic encoding and retrieval (*Rugg and Vilberg, 2013*) and constructive memory-related tasks such as imagining fictitious scenes and future events (*Addis et al., 2007*; *Hassabis and Maguire, 2007*; *Hassabis and Maguire, 2007*; *Rugg and Vilberg, 2013*; *Schacter and Addis, 2007*; *Schacter and Addis, 2007*). Our design relies on an event-level correspondence between the encoding (viewing) and verbal recall of movie scenes. Previous research has localized modality-independent representations of movie scenes (*Zadbood et al., 2017*) and their similarity during encoding and recall (*Chen et al., 2017*) to the DMN. These characteristics make this network a good candidate to contribute to memory updating—a constructive process in which new information is integrated into past event memories in service of better guiding behavior. Our findings support this idea by showing the shift in neural representations during updated recall in a subset of regions in this network.

At encoding, a widespread set of areas including dorsal and lateral PFC, left precuneus, left retrosplenial cortex, and left angular gyrus had differentiable neural patterns across the two interpretations of the movie. These results are consistent with previous work that showed the time course of brain responses in DMN distinguishes between groups when participants are prompted to take two different perspectives before listening to an audio story (*Yeshurun et al., 2017*). We extend these results to an audiovisual movie, and provide evidence that interpretative perspective is also encoded in spatially distributed neural response patterns for narrative events, averaged across minutes-long scenes. Interestingly, the difference in neural responses measured by Yeshurun and colleagues was not significant between the two perspectives of the story in the ventral portion of mPFC. Similarly, vmPFC ROIs did not exhibit a significant difference between the *Doctor* and *Ghost* representations during the encoding phase in our experiment. Previous research has implicated mPFC in processing schematic information and integration of new information into prior knowledge (*Gilboa and Marlatte, 2017*; *Schlichting and Preston, 2017*; *van Kesteren et al., 2012*). Using naturalistic clips as schematic events, it has been shown that response patterns in mPFC are particularly dependent on intact and predictable schemas (*Baldassano et al., 2018*). Together, these results suggest that our manipulation (*Doctor* and *Ghost* interpretations) may not have substantially altered the schemas that participants were using during movie-viewing (e.g. during a restaurant scene, participants will need to use their 'restaurant' schema to interpret it, regardless of whether the doctor is alive or a ghost)—although we interpret these null results with caution.

Even though groups had different knowledge/perspectives during encoding, we found higher pattern similarity across groups if they had similar twist knowledge during recall in vmPFC, right precuneus, and parts of temporal cortex. Previous findings suggest mPFC is involved in not just encoding but retrieval of memories in relation to prior knowledge (*Brod et al., 2015*; *van Kesteren et al., 2010*) and retrieval of overlapping representations to support integration and organization of related memories (*Tompary and Davachi, 2017*). Our observations during recall fit with these findings and suggest that shifting toward a more similar perspective during recall leads to higher neural similarity in mPFC. However, during encoding, we did not observe a significant pattern correlation between groups that held the same interpretation of the movie. Furthermore, vmPFC was significant in our interaction analysis (*Figure 2C*), indicating that the similarity structure of vmPFC patterns across conditions was significantly different at encoding versus retrieval. Together, these results suggest vmPFC is differently implicated in encoding and recall of story-specific representations during processing of naturalistic events. In addition to mPFC, right precuneus and parts of temporal cortex exhibited significantly higher pattern similarity in the 'twist' and 'spoiled' groups who recalled the movie with the same

interpretation. Precuneus is a core region in the posterior medial network, which is hypothesized to be involved in constructing and applying situation models (*Ranganath and Ritchey, 2012*). Our findings support a role for precuneus in deploying interpretation-specific situation models when retrieving event memories. In particular, we suggest that the posterior medial network may encode a shift in the situation model of the 'twist' group in order to accommodate the new *Ghost* interpretation.

We performed two targeted analyses to look for evidence of memory updating across encoding and recall: the interaction analysis (*Figure 2C*) and the encoding-recall analysis (*Figure 3*). We hypothesized that a shift in direction of pISC difference would occur when neural representations during recall in the 'twist' group start to reflect the *Ghost* interpretation. The interaction analysis probed this shift indirectly by taking into account the effects of both encoding-encoding and recall-recall analyses. Unlike the interaction analysis, in the encoding-recall analysis, we *directly* compared neural event representations during encoding and recall. Interestingly, all regions exhibiting an effect across the two encoding-recall analyses, excluding left anterior temporal cortex, were present in the interaction results. Among these regions, the left angular gyrus/TPJ exhibited an effect across all three analyses. As a core hub in the mentalizing network, temporo-parietal cortex has been implicated in theory of mind through perspective-taking, rationalizing the mental state of someone else, and modeling the attentional state of others (*Frith and Frith, 2006*; *Guterstam et al., 2021*; *Saxe and Kanwisher, 2003*). The motivations behind some actions of the main character in the movie heavily depend on whether the viewer perceives them as a Doctor or a Ghost, and participants may focus on this during both encoding and recall. We speculate that neural event representations in AG/TPJ in the current experiment may be related to mentalizing about the main character's actions. Under this interpretation, the updated event representations during recall following the twist would be more closely aligned to the 'spoiled' encoding representations, as a consequence of memory updating in the 'twist' group.

Our findings are consistent with the view that DMN synthesizes incoming information with one's prior beliefs and memories (*Yeshurun et al., 2021*). We add to this framework by providing evidence for the involvement of DMN regions in updating prior beliefs in light of new knowledge. Across our different encoding and recall analyses, we observe memory updating effects in a varied subset of DMN regions that do not cleanly map onto a specific subsystem of DMN (*Robin and Moscovitch, 2017*; *Ranganath and Ritchey, 2012*; *Ritchey and Cooper, 2020*). Rather than being divergent, these results might be reflecting inherent differences between the processes of encoding and recall of naturalistic events. It has been proposed that neural representations corresponding to encoding of events are systematically transformed during recall of those events (*Chen et al., 2017*; *Favila et al., 2020*; *Musz and Chen, 2022*). While we provide evidence for reinstatement of memories in DMN, our findings also support a transformation of neural representation during recall, as encoding-recall results were weaker in some areas than recall-recall findings. This transformation could affect how different regions and sub-systems of DMN represent memories, and suggests that the concerted activity of multiple subsystems and neural mechanisms might be at play during encoding, recall and successful updating of naturalistic event memories.

While our main goal in this paper was to examine how neural representations of naturalistic events change in the DMN, we also examined visual and somatosensory networks. Aside from the encoding-encoding analysis in which some visual and somatosensory regions showed stronger similarity between two groups with the same interpretation of the movie, we did not find any regions with significant effects in these two networks in the other analyses. Unlike the recall phase where each participant has their unique utterance with their own choice of words and concepts to describe the movie, the encoding (move-watching) stimulus is identical across all groups. Therefore, the effects observed during encoding-encoding analysis in sensory regions could reflect similarity in perception of the movie guided by similar attentional state while watching scenes with the same interpretation (e.g. similarity in gaze location, paying attention to certain dialogues, or small body movements while watching the movie with the same Doctor or Ghost interpretations). In our whole brain analysis, these regions did not have significant interaction effects, which suggests that the effects were isolated to encoding. In the whole-brain analysis, we also observed a significant encoding-encoding and interaction effects in anterior cingulate cortex, as well as recall-recall and interaction effects in dlPFC. These results suggest that both the 'spoiled' manipulation and the 'twist' may recruit top-down control and conflict monitoring processes during naturalistic viewing and recall.

Our findings provide further insight into the functional role of the DMN. However, these results have been obtained using only one movie. While naturalistic paradigms better capture the complexity of real life and provide greater ecological generalizability than highly-controlled experimental stimuli and tasks (*Nastase et al., 2020*), they are still limited by the properties of the particular naturalistic stimulus used. For example, this movie—including the twist itself—hinges on suspension of disbelief about the existence of ghosts. Future work is needed to extend our findings about updating event memories to a broader class of naturalistic stimuli: for example, movies with different kinds of (non-supernatural) plot twists, spoken stories with twist endings, or using autobiographical real-life situations where new information (e.g. discovering a longtime friend has lied about something important) triggers re-evaluation of the past (e.g. reinterpreting their friend's previous actions). Moreover, our current method relies on averaging spatially coarse activity patterns across subjects (and time points within an event). Future extensions of this work may benefit from using functional alignment methods (*Haxby et al., 2020*; *Chen et al., 2015*) to capture more fine-grained event representations which are shared across participants.

During recall, many participants recounted both the old and new interpretations (*Ghost* and *Doctor*) of movie scenes. This behavior indicated that they maintained both representations in parallel (possibly competing), rather than overwriting the old representation with new information. The simultaneous presence of these representations poses an interesting theoretical question for future studies: When does updating the memory cause us to lose traces of the old interpretation, and when do the old and new interpretations end up co-existing in memory? Previous studies have shown that old and new memory traces are simultaneously reactivated in the brain, leading to competition (e.g. *Kuhl et al., 2012*), and this competition can trigger learning processes that resolve the competition; for example, by weakening one of the memories or by restructuring the memories so they can coexist (*Ritvo et al., 2019*). Understanding how competition between interpretations plays out over time is an important topic for future work; existing research on memory revaluation suggests that updating may be a temporally-extended process driven by successive replays of the new information, rather than taking place all at once (see, e.g., *Momennejad et al., 2018*). In clinical settings, methods inspired by reconsolidation and memory updating are extensively used to treat maladaptive memories (*Phelps and Hofmann, 2019*). In these clinical contexts, it will be especially important to understand the factors that influence the 'end state' of this competition between interpretations (in terms of our study: who ends up fully adopting the *Ghost* interpretation and who ends up with lingering traces of the *Doctor* interpretation).

In summary, our findings show that in a movie with a dramatic twist ending, the new information introduced by the twist causes a new (*Ghost*) interpretation of past events to take root in participants' brains. Overall, these results highlight the importance of DMN regions in updating naturalistic memories and suggest new approaches to studying real-world memory modification in both experimental and clinical treatment settings.

## Materials and methods

### Stimuli

The stimuli consisted of three edited versions of 'The Sixth Sense' (M. Night Shyamalan, 1999) movie. The movie depicts the story of a child psychologist treating a young boy who claims he can see and speak with dead people. In the film's ending scene, however, it is revealed that the psychologist died prior to the events of the movie and has actually been one of the ghosts the boy was seeing all along. Three different edited versions of the movie were created for the experiment. The first version was a~60 min shortened movie including the final scene with the big reveal followed by a text on the screen describing the twist to ensure all participants in the 'twist' group fully understood the twist information. The second version was identical to the first version, but a spoiler was presented as text on screen early in the movie (the 'spoiled' group). In the third version, the final scene was cut out and the movie ended at a point where it appeared that the doctor successfully completed the treatment and therefore did not raise any suspicion about the twist in participants who watched this version ('no-twist' group). Eighteen scenes were selected to be included in the cued recall task (see the section on timestamping and scene selection below). For each of these scenes, a short clip from

the beginning of that scene (lasting from 5 to 36 s. Mean = 12.9 s) was used as a retrieval cue for the scene during the recall task.

## Participants

Sixty-six right-handed, native English speakers (ages 18–24, average = 20, 21 males) were scanned in the experiment. Our sample size was decided based on our previous work in which we captured scene-specific pattern similarity across encoding, recall, and listening (18 subjects per group in *Zadbood et al., 2017*, 17 subjects in *Chen et al., 2017*) and differences in brain response while listening to the same story with different perspectives (20 subjects per group in *Yeshurun et al., 2017*). None of the participants had previously seen The Sixth Sense in full or in part, which was confirmed through an online questionnaire before the session. However, because the movie is well-known and frequently referenced in popular culture, participants with some knowledge about the twist (e.g. knowing that this is a movie about ghosts and the main character is actually dead) were admitted to the 'spoiled' group (see *Experimental design*) in order to facilitate data collection. In the post-scan questionnaire, two participants reported guessing the twist while watching the movie and their data were excluded. One participant did not understand the twist after watching the final scene and receiving the text explanation, so their data were omitted as well. Six participants were excluded due to large head motion (spikes of framewise displacement >4 mm). The data of the remaining fifty-seven participants were used in the analyses. All participants provided written informed consent prior to the experiment and received information about the conditions of the experiment and their rights. The experiment protocol and the consent forms were approved by the Institutional Review Board of Princeton University.

## Experimental design

Participants were pseudo-randomly divided into three groups: the 'twist' group (N=19) watched a 60 min audio-visual edition of The Sixth Sense movie, including the twist at the end while undergoing fMRI scanning. The 'spoiled' group (N=18) watched a spoiled version of the movie (see Stimuli). The 'no-twist' group (N=20) watched a 55 min version of the movie with no twist scene (*Figure 3B*). Participants were instructed to watch the movie naturally and attentively, as there will be a task related to the movie content after watching. However, no specific information about the upcoming recall task was revealed. After the movie, participants performed a verbal cued recall task. During the cued recall task, participants watched short clips from 18 scenes of the movie. After each clip, they were asked to freely describe the events of that particular scene and to provide the most accurate interpretation of the scene given all the information they have gathered throughout watching. The instructions were identical for all three groups. The movie cue and recall were separated by 14 s, which ended as a countdown on the screen. The recall task was self-paced and participants pressed a button to continue to the next scene after each recall. After scanning, participants filled out a questionnaire about their experience in the scanner, including information about the movie and recall tasks and whether they guessed the twist in the middle of the movie (and if yes in which scene). All participants rated the movie as engaging. Participants in the 'no-twist' group were debriefed about the real ending of the movie before leaving the facility.

## Scanning procedure

The scanning session began with an anatomical scan. Participants watched the movie and read the instructions through a mirror mounted to the head coil which reflected a rear screen. The main screen was located at the back of the magnet bore and the movie was projected on the screen via an LCD projector. MR-safe, in-ear headphones were used for the movie audio. Eye-tracking was set up to monitor participants during the scans in real-time and ensure they stayed awake and attentive during the experiment. The movie and recall stimuli were presented using the Psychophysics Toolbox in MATLAB (Mathworks), which enabled coordinating the onset of the stimuli (movie and recall cues) with data acquisition. The volume level of the movie was adjusted separately for each participant using a sample clip to assure a clear and comfortable audio signal. Recall speech was recorded during the fMRI scan using a customized MR-compatible recording system (FOMRI II; Optoacoustics Ltd). The MR recording system used two orthogonally-oriented optical microphones. The reference microphone captures the background noise, and the source microphone captures both background noise

and the speaker's speech (signal). A dual-adaptive filter subtracted the reference input from the source channel using a least mean square approach. To achieve an optimal subtraction, the reference signal was adaptively filtered so the filter gains are learned continuously from the residual signal and the reference input. To prevent divergence of the filter when speech was present, a voice activity detector was integrated into the algorithm. A speech enhancement spectral filtering algorithm further preprocessed the speech output to achieve a real-time speech enhancement. Audio recordings were further cleaned using noise removal software (Adobe Audition). The output recall recordings were fully comprehensible. A response box was used to collect the participants' manual button-presses during the recall task. Participants were instructed to press a button when they finished the recall of a scene to proceed with the task. In five participants, the recall scans were stopped due to problems in pressing the buttons (or just by mistake) and were resumed after they received feedback and further instructions. In these cases, the recalls were resumed starting with the next scene. In three participants the recall scan was stopped after the first scene and in one participant before the last two scenes. In one participant the scan stopped and resumed in the middle of the recall task.

## MRI acquisition

MRI data were collected on a 3T full-body scanner (Siemens Prisma) with a 64-channel head coil. Functional images were acquired using an interleaved multiband EPI sequence (TR = 1500ms, TE 33ms, flip angle 80 degrees, whole-brain coverage, 2 mm slice thickness, FOV 192 mm$^2$, SMS = 4). Anatomical images were acquired using a T1-weighted magnetization-prepared rapid-acquisition gradient echo (MPRAGE) pulse sequence (1 mm$^3$ resolution). Anatomical images were acquired in a 6-min scan before the functional scans with no stimulus on the screen. Field maps were collected for B0 correction at the end of the recall run.

## Preprocessing

Preprocessing was performed using fMRIPrep, version stable 1.0.11 (**Esteban et al., 2019**, RRID:SCR_016216), a Nipype (**Gorgolewski et al., 2011**, RRID:SCR_002502) based tool. Each T1w (T1-weighted) volume was corrected for INU (intensity non-uniformity) using N4BiasFieldCorrection v2.1.0 (**Tustison et al., 2010**) and skull-stripped using antsBrainExtraction.sh v2.1.0 (using the OASIS template). Spatial normalization to the ICBM 152 Nonlinear Asymmetrical template version 2009c (**Fonov et al., 2009**, RRID:SCR_008796) was performed through nonlinear registration with the antsRegistration tool of ANTs v2.1.0 (**Avants et al., 2008**, RRID:SCR_004757), using brain-extracted versions of both T1w volume and template.

Functional data were motion corrected using mcflirt (FSL v5.0.9, **Jenkinson et al., 2002**). "Fieldmapless" distortion correction was performed by co-registering the functional image to the same-subject T1w image with intensity inverted (**Wang et al., 2017**) constrained with an average fieldmap template (**Treiber et al., 2016**), implemented with antsRegistration (ANTs). This was followed by co-registration to the corresponding T1w using boundary-based registration (**Greve and Fischl, 2009**) with six degrees of freedom, using flirt (FSL). Motion correcting transformations, field distortion correcting warp, BOLD-to-T1w transformation and T1w-to-template (MNI) warp were concatenated and applied in a single step using antsApplyTransforms (ANTs v2.1.0) using Lanczos interpolation.Frame-wise displacement (**Power et al., 2014**) was calculated for each functional run using the implementation of Nipype.

Then, the datasets were adaptively smoothed using AFNI's 3dBlurToFWHM to reach 7 mm global smoothness (**Cox, 1996**). Note that the 7 mm reported smoothness is the *global smoothness*, which is the 'final' smoothness of the images given their original, intrinsic smoothness and the applied smoothing. In other words, we did not apply an additional 7 mm smoothing kernel to the data; rather, we iteratively smoothed the data until a 7 mm global smoothness was attained (using AFNI's 3dBlurToFWHM). If the initial smoothness of the raw data was roughly 2 mm, this would be similar to applying a 5 mm smoothing kernel. This amount of smoothing is comparable to previous papers using similar intersubject pattern similarity methods to compare event-level representations during encoding and recall (**Chen et al., 2017**; **Zadbood et al., 2017**). AFNI's 3dTproject was used to regress out confound variables comprising head motion (6 motion parameters and their temporal derivatives), second-order polynomial detrending variables, and high-pass filtering (140 s cutoff). De-spiking and subsequent analyses were conducted using custom MATLAB scripts (see Code Accessibility). The movie data

were acquired in a single run and the time series were z-scored across the entire run prior to further analysis. Inspection of the recall data revealed a dramatic difference in mean signal intensity between the audiovisual movie cues and the verbal recall sections during the cued-recall task. To account for this, we used the least-squares-separate (LSS) method (*Mumford et al., 2012*) implemented by AFNI's 3dLSS to model the recall data. In this method each verbal recall section was modeled independently of both the other recall scenes and the preceding movie cue. Regression coefficients (beta values) obtained by this method (one beta value per scene) were used in the main analyses. In four participants where the recall scan was split due to button-press issues, the smaller section of the recall only included 1–2 scenes. These scans were too short to be modeled using LSS and the data for these scenes were ignored. All analyses were performed in volume space. The results were projected onto the surface for visualization using Connectome Workbench.

## Atlas and ROI definitions

Whole brain ROI analysis was performed on a set of 100 ROIs grouped into seven networks based on functional connectivity during rest (*Schaefer et al., 2018*). Twenty-four of these ROIs labeled as 'DMN' were used in the main analysis.

## Timestamping and scene selection

The movie was time-stamped by an independent rater naive to the purpose and design of the experiment to identify the main scenes of the movie. All of the movie scenes with clear scene boundaries (N=18) were selected to be used in the cued-recall task. Prior to running the fMRI experiment, our evaluation of movie scenes suggested that, in some scenes, knowing the twist information would more dramatically change the interpretation of the scenes (we called them 'critical scenes'). However, we also thought it was possible that the twist would affect recall of other scenes in the movie; for this reason, we decided to include all 18 movie scenes in the cued-recall task, rather than limiting ourselves to the critical scenes. Very short snippets from the beginning of these scenes were used as cues in the recall task. To determine the exact number of 'critical scenes', a group of five raters (AZ from the authors and four independent raters naïve to the purpose of the experiment) watched the movie and rated all 18 scenes in terms of how much the twist information might change the interpretation of these scenes ('twist influence'). They were instructed to rate each scene on a scale of 1–5 (1=Interpretation does not change at all, 2=Interpretation is mildly changed, 3=Interpretation is moderately changed, 4=Interpretation is strongly changed, 5=Interpretation is very strongly changed). Six scenes scored 4 or higher ('Interpretation is strongly changed')—these critical scenes were selected for the main neural analyses. There was 100% agreement between the top 6 scenes scored by the rater from our group and the top 6 scenes selected using the average of ratings of our four independent raters. In the independent analysis of the recall behavior data, this same set of 6 scenes scored highest in twist score (described in the next section) which indicates a match between expected and perceived effect of twist information on recall behavior. Scene number one, in which the doctor and child meet for the first time was scored ~3 (Interpretation is moderately changed) but showed a high twist score in the behavioral recall analysis. This scene was the first time participants recalled the doctor after the main reveal (watching the twist) and given its high twist score, the recall and possibly the corresponding neural patterns appeared to be more strongly affected by the twist information. Therefore, we added this scene as a seventh critical scene to be used in the main neural analyses.

## Behavioral analysis

The recall data were transcribed from speech to text and subject numbers (and group information) were removed. The same four independent raters who watched the movie and rated the 'twist influence' in the previous section read the recall data scene by scene. They rated each scene for all subjects, while the order of scenes across subjects was shuffled and there was no information indicating to which experimental group the scene belonged. They were asked to report a score for each scene based on the 'ghostness' or 'doctorness' of the depiction of the main character in that scene. The scores were from 1 to 5 (1=Purely reflects the *Doctor* interpretation, 2=More strongly reflects the *Doctor* interpretation, 3=Balanced between *Doctor* and *Ghost* interpretation, 4=More strongly reflects the *Ghost* interpretation, 5=Purely reflects the *Ghost* interpretation). Raters showed strong agreement on their scoring (pairwise correlations between raters' scores ranged from $r$=0.84, p=6.6

$\times\ 10^{-18}$ to $r$=0.97, p=7.5 $\times\ 10^{-42}$). Scores for each scene were averaged across 4 raters and were used as the twist score in the main analyses. Two separate raters scored the recall data based on the details and accuracy of recall irrespective of the twist information. Scores provided by these raters were averaged and used as the 'memory score'.

The average length of scenes in the 55-min movie was 2 min and 10 s (sd = 1:59, median = 1:56, min = 00:26, max = 5:56). For the recalls, in the 'spoiled' group the average recall time per scene was 39.4 s (sd = 13.2 s, min = 14 s, max = 67 s) for a total average of 713 s of recall time. In the 'twist' group, the average recall time per scene was 38.5 s (sd = 13 s, min = 17 s, max = 69 s) for a total average of 698 s of recall time. In the 'no-twist' group, the average recall time per scene was 37.75 s (sd = 19.8 s, min = 8 s, max = 73 s) for a total average of 642 s of recall time. No significant differences were observed between average recall time per scene or overall recall time across any two groups (according to t-tests).

## Intersubject pattern correlation (pISC) analysis

The multivariate analysis of the data was performed by measuring the similarity between the spatial patterns of brain response in each ROI. To obtain this measure, first the time series of brain responses to the movie in each subject/ROI was averaged within each of the seven critical scenes. This method has been used to study scene-specific patterns of brain activity in previous studies (*Chen et al., 2017*; *Zadbood et al., 2017*). Averaging the time series within each scene resulted in seven spatially distributed patterns of brain activity in each ROI. For the recall phase, the beta values extracted via LSS modeling were used, similarly providing 7 activity patterns in each ROI. All pattern similarity analyses were performed between subjects to capitalize on the between-group design of the experiment (*Nastase et al., 2019*). For the encoding phase, the patterns of brain activity in each subject from the 'twist' group were correlated (Pearson correlation) with the average of activity patterns for the 'spoiled' group in corresponding scenes and averaged across scenes. The same procedure was performed to compare the 'twist' and 'no-twist' groups which resulted in two correlation values assigned to each subject in the 'twist' group. All correlation values were Fisher transformed prior to further analysis (*Fisher, 1915*). In each ROI, the difference between these two comparisons was calculated and averaged across participants (difference r values depicted on each map). To determine statistical significance, we compared these two sets of values using a non-parametric paired t-test by shuffling the sign of difference values across subjects 1000 times and calculating a p-value for the observed difference based on this null distribution (one-tailed). p Values were corrected for multiple comparisons across DMN ROIs by controlling the false discovery rate (FDR) at p<0.05 (*Benjamini and Hochberg, 1995*). The same procedure was performed in the recall and encoding-recall analysis except for two differences in the encoding-recall analysis: during the analysis to compare 'twist' and 'no-twist' recall with 'spoiled' encoding (*Figure 3A*), an independent sample non-parametric t test was performed by shuffling the group labels 1000 times and calculating the difference between the two permuted groups at each iteration to create the null distribution. To compare the 'twist' recall with the 'twist' encoding (*Figure 3B*), each subject's recall was compared to the average of the rest of the group's encoding to ensure all comparisons were made across subjects. To match the number of subjects in the encoding groups, one subject was randomly dropped from the encoding set in each iteration when comparing 'twist' recall to 'spoiled' encoding.

The interaction analysis assessed whether neural patterns in the 'wist' group were relatively more similar to the 'spoiled' (vs. 'no-twist') group at recall (vs. encoding), and was computed as follows:

$$\textbf{interaction index (r)} = (\textit{movie-no-twist}\ \text{vs.}\ \textit{movie-twist}) - (\textit{movie-spoiled}\ \text{vs.}\ \textit{movie-twist}) -$$

$$\left[ (\textit{recall-no-twist}\ \text{vs.}\ \textit{recall-twist}) - (\textit{recall-spoiled}\ \text{vs.}\ \textit{recall-twist}) \right]$$

The same pISC procedures were performed in the regions of the visual and somatosensory cortices (*Appendix 1—figure 2*), as well as across the whole brain (*Appendix 1—figure 3*). All regions were extracted from the 100-parcel Shaeffer where each region is assigned to one of seven resting-state networks (*Schaefer et al., 2018*).

In the scene content analyses (*Figure 4*), interaction index was calculated using different subsets of scenes including *critical scenes* (N=7), *non-critical scenes* (N=11) and *all scenes* (N=18) in all regions within the DMN. Other than the scene selection, the rest of the procedure was identical to the pISC analysis described above and used in the main analyses (*Figure 2*). Interaction indices across ROIs

from the 'critical scene' condition were compared with the ones from all scene and non-critical scenes using a paired t-test. To examine scene specificity, scene labels were shuffled within each subject and the pISC analysis was repeated 1000 times on the new scene orders to create the distribution depicted as 'shuffled scenes' on *Figure 4*. For each ROI, the number of interaction indices in this pool of null values that were larger than the original interaction indices were used to calculate p-values. p-Values were corrected for multiple tests (across DMN regions) by controlling the false discovery rate (FDR) at q<0.05.

To ensure our results (*Figures 2–3*) were not driven by overall activation differences across the groups ('spoiled', 'twist', 'no-twist'), we performed a univariate analysis in each ROI. For each participant in the movie group, we calculated the regional-average response magnitude in each ROI. The same procedure was done for beta values obtained using the GLM during recall. This yielded a two conditions by three groups table of univariate activity magnitudes per ROI. We performed an ANOVA with condition as a within-subject factor and group as a between-subject factor. This analysis yielded no significant effect of group or interaction of group and condition in any ROIs. Two ROIs exhibited a significant effect of group and one ROI showed a significant interaction prior to correction for multiple tests, but these values did not survive correction (*Appendix 1—table 1*).

To ensure that our results were not biased due to any systematic differences in the noise level of neural activity patterns between the groups (*spoiled*, *twist*, *no-twist*), we calculated the pISC within each group by correlating each subject's pattern with the average pattern from the rest of the subjects in that group. We performed this procedure for the movie and recall conditions separately in each of the 15 ROIs that showed any significant effect in any of the reported analyses. We then submitted all the correlation values across subjects to an ANOVA including all groups, conditions, and ROIs. As expected, we did not find any main effect of group or an interaction of group with condition or ROI.

In the analysis to identify the relationship between the neural and behavioral signature of memory update (*Appendix 1—figure 1*), the neural data were obtained by computing (r*ecall-twist* vs. *movie-spoiled*) – (r*ecall-twist* vs. *movie-twist*), as mentioned above and described in the results section. However, the difference values were not averaged and were correlated with the twist score across participants.

In the analysis to compare the behavioral and neural responses across scenes (*Appendix 1—figure 4*), we performed the same pISC procedure as in the recall-recall analysis (*Figure 2B*) and encoding-recall analysis (*Figure 3B*) on all scenes. However, this time we did not average the scene responses. Instead, we averaged the pISC difference value for each scene across participants. Similarly, we averaged participants' behavioral twist scores for each scene. We then computed the Pearson correlation between the vectors of pISC differences and behavioral twist scores (18 values in each vector, equal to the number of scenes). We repeated this procedure in each DMN region and plotted the strength of these correlations on the brain (*Appendix 1—figure 4*, left panels). To better understand the role of 'critical scenes' in this relationship, those scenes were marked as blue on the scatter plots (*Appendix 1—figure 4*, left panels).

## Code and data accessibility

Code available at: https://github.com/azadbood/sixthsense, (copy archived at swh:1:rev:889b89f-8201d7a28b9dc2b44dda211f05a218c0d; *Asieh, 2022*). Data available at: https://doi.org/10.18112/openneuro.ds004359.v1.0.0.

## Acknowledgements

This work was supported by the National Institutes of Health (NIMH R01 MH12357 awarded to UH and KAN and DP1 HD091948 awarded to UH). We thank Savannah Born for help in analyzing the behavioral data, Christopher Honey for helpful comments on data analysis, and Liat Hasenfratz for help with execution of the study.

# Additional information

## Funding

| Funder | Grant reference number | Author |
| --- | --- | --- |
| National Institute of Mental Health | R01 MH12357 | Kenneth A Norman Uri Hasson |
| National Institutes of Health | DP1 HD091948 | Uri Hasson |

The funders had no role in study design, data collection and interpretation, or the decision to submit the work for publication.

## Author contributions

Asieh Zadbood, Conceptualization, Data curation, Software, Formal analysis, Investigation, Visualization, Methodology, Writing - original draft, Project administration, Writing - review and editing; Samuel Nastase, Software, Formal analysis, Investigation, Methodology, Writing - review and editing; Janice Chen, Conceptualization, Supervision, Methodology, Writing - review and editing; Kenneth A Norman, Uri Hasson, Conceptualization, Supervision, Funding acquisition, Methodology, Writing - review and editing

## Author ORCIDs

Asieh Zadbood http://orcid.org/0000-0002-9098-0510
Samuel Nastase http://orcid.org/0000-0001-7013-5275
Kenneth A Norman http://orcid.org/0000-0002-5887-9682

## Ethics

Human subjects: All participants provided written informed consent prior to the experiment and received information about the conditions of the experiment and their rights. The experiment protocol and the consent forms were approved by the Institutional Review Board of Princeton University (protocol number 7883).

## Decision letter and Author response

Decision letter https://doi.org/10.7554/eLife.79045.sa1
Author response https://doi.org/10.7554/eLife.79045.sa2

# Additional files

## Supplementary files

• MDAR checklist

## Data availability

Code available at: https://github.com/azadbood/sixthsense, (copy archived at swh:1:rev:113b-3203722573033b5cc4535f6523c1fca5c1d7) Data available at: https://doi.org/10.18112/openneuro.ds004359.v1.0.0.

The following dataset was generated:

| Author(s) | Year | Dataset title | Dataset URL | Database and Identifier |
| --- | --- | --- | --- | --- |
| Zadbood A, Nastase SA, Chen J, Norman KA, Hasson U | 2022 | SixthSense | https://doi.org/10.18112/openneuro.ds004359.v1.0.0 | OpenNeuro, 10.18112/openneuro.ds004359.v1.0.0 |

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

# Appendix 1

Relationship between the behavioral (twist score) and neural (R$_G$ to M$_G$ > R$_G$ to M$_D$) measures of memory update across subjects

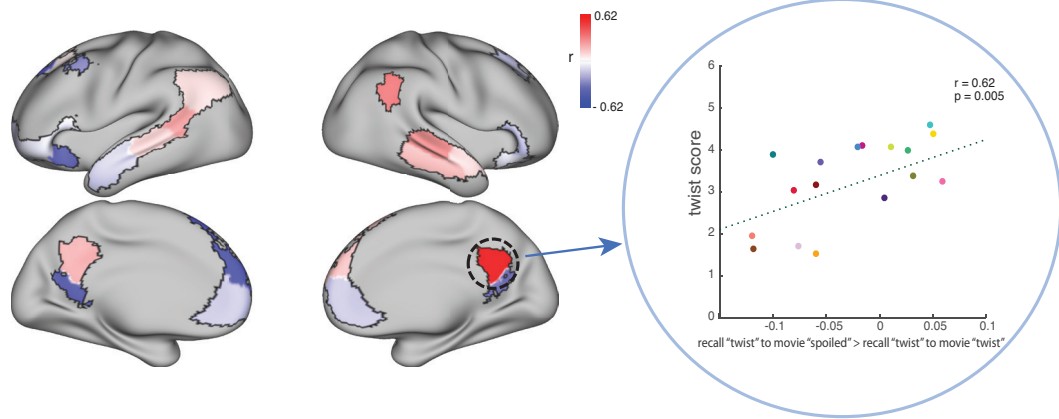

**Appendix 1—figure 1.** The relationship between the behavioral (twist score) and neural (recall 'twist' to movie 'spoiled'>recall 'twist' to movie 'twist') measures of memory update in each DMN ROI. The panel on the right depicts the correlation in the precuneus. Each dot is a participant in the 'twist' group (N=19). Note that the example at right was selected post-hoc for high correlation and is not significant after correction for multiple tests.

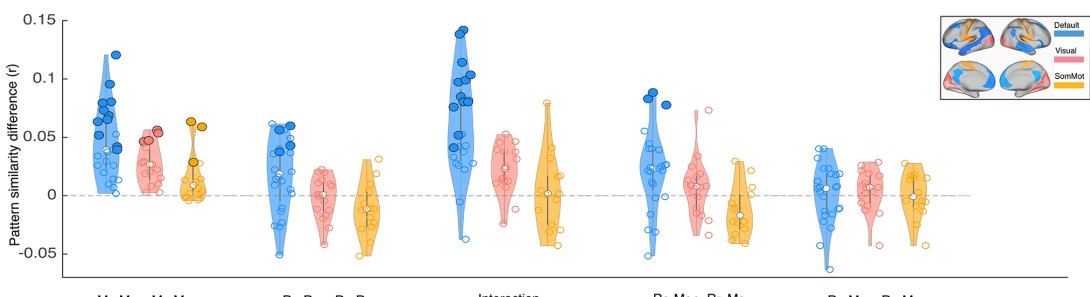

**Appendix 1—figure 2.** Depiction of pattern similarity differences calculated in the main analyses (*Figures 2–3*) in three resting state networks. Blue graphs depict the results in the default mode network. Light red represents the visual network, and yellow shows the somatosensory network. Each dot on the violin graphs shows the effect size in one of the regions on interest in the corresponding network. White dots depict median values. Significance levels have been calculated for each ROI separately (the same as the main analysis). Bold dots depict ROIs with p<0.01 (uncorrected). Labels on x axis: D subscript denotes *Doctor* interpretation and G subscript denotes *Ghost* interpretation. M$_D$–M$_D$ >M$_D$–M$_G$ denotes greater pISC when comparing naive encoding in 'twist' group (M$_D$) to naive encoding in 'no-twist' group (M$_D$) than when comparing naive encoding in 'twist' group (M$_D$) to spoiled encoding in 'spoiled' group (M$_G$) (similar comparison as in *Figure 2A* and *Appendix 1—figure 3A*). R$_G$–R$_G$ >R$_G$–R$_D$ denotes greater pISC when comparing recall in 'twist' group (R$_G$) to recall in 'spoiled' group (R$_G$) than when comparing recall in 'twist' group (R$_G$) to naive recall in 'no-twist' group (R$_D$) (similar comparison as in *Figure 2B* and *Appendix 1—figure 3B*). Interaction corresponds to the same analysis in *Figure 2C* and Figure *Appendix 1—figure 3C*. R$_G$–M$_G$ >R$_D$–M$_G$ denotes greater pISC when comparing updated recall in 'twist' group (R$_G$) to spoiled encoding in 'spoiled' group (M$_G$) than when comparing naive recalls in 'no-twist' group (R$_D$) to spoiled encoding in 'spoiled' group (M$_G$) (similar comparison as in *Figure 3A* and *Appendix 1—figure 3D*). R$_G$–M$_G$ >R$_G$–M$_D$ denotes greater pISC when comparing updated recall in 'twist' group (R$_G$) and spoiled encoding in *spoiled* group (M$_G$) than when comparing updated recall in 'twist' group (R$_G$) and naive encoding in 'twist' group (M$_D$) (similar comparison as in *Figure 3B* and *Appendix 1—figure 3E*).

A) movie-twist (M$_D$) to movie-no-twist (M$_D$) > movie-twist (M$_D$) to movie-spoiled (M$_G$)

B) recall-twist (R$_G$) to recall-spoiled (R$_G$) > recall-twist (R$_G$) to recall-no-twist (R$_D$)

C) interaction

D) recall-twist (R$_G$) to movie-spoiled (M$_G$) > recall-no-twist (R$_D$) to movie-spoiled (M$_G$)

E) recall-twist (R$_G$) to movie-spoiled (M$_G$) > recall-twist (R$_G$) to movie-twist (M

p < 0.01 NOT corrected

**Appendix 1—figure 3.** Depiction of the same set of results as in *Figure 2* (upper row) and *Figure 3* (lower row) in the whole brain (not restricted to DMN). The maps show ROIs with p<0.01 calculated by nonparametric randomization test without correction (areas missing on these maps compared to the original maps had p values greater than 0.01). (**A**) Areas with significantly greater intersubject pattern correlation between groups who encoded the movie with the same interpretation (*Doctor*). (**B**) Areas with significantly greater intersubject pattern correlation between groups who recalled the movie with the same interpretation (*Ghost*). (**C**) Areas with a significant interaction effect, indicating a change in interpretation between encoding and recall (see 'Pattern similarity analysis' in Methods). (**D**) Areas where intersubject pattern correlations are significantly greater when comparing updated recall (R$_G$) to spoiled encoding (M$_G$) than when comparing naive recalls (R$_D$) to spoiled encoding (M$_G$). (**E**) Areas where intersubject pattern correlations between updated recall (R$_G$) and spoiled encoding (M$_G$) are greater than between updated recall (R$_G$) and naive encoding (M$_D$).

A) Relationship between the behavioral (twist score) and neural (R_G to R_G > R_G to R_D) measures of memory updating **across scenes**

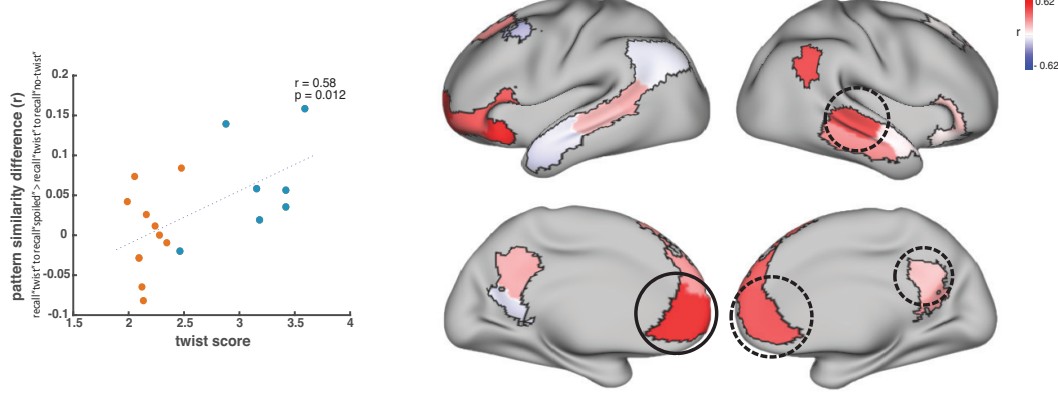

B) Relationship between the behavioral (twist score) and neural (R_G to M_G > R_G to M_D) measures of memory updating **across scenes**

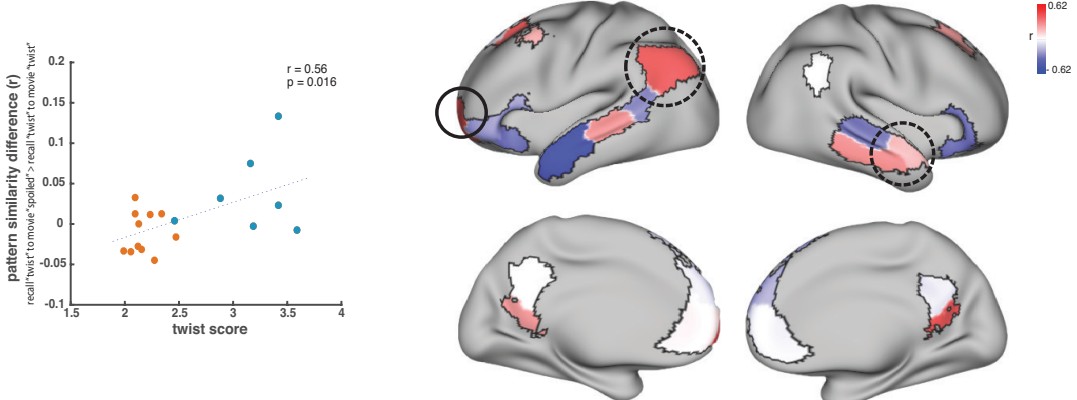

**Appendix 1—figure 4.** Relationship across scenes between our behavioral measure of memory updating (twist score) and neural measures of memory updating (pISC difference). (**A**) shows the results when we operationalize memory updating based on the difference between pISC for recall 'twist' to recall 'spoiled' and pISC for recall 'twist' to recall 'no-twist'. Correlation values for each DMN region are shown as colors on the map. Red shade indicates higher correlation. Circled areas highlight the regions that showed significant effects in the main recall-recall analysis across subjects (**Figure 2B**), where pISC for recall 'twist' to recall 'spoiled' was greater than pISC for recall 'twist' to recall 'no-twist'. The area with the highest correlation among these areas (which is the highest correlation in the entire map as well) is highlighted with a black continuous-line circle. The scatter plot on the left depicts the correlation in this area. 'Critical scenes' that were used in the main analyses (**Figures 2–3**) are shown in blue on the scatter plot. The rest of the scenes are shown in red. Correlation was calculated across all 18 scenes. (**B**) shows the results when we operationalize memory updating based on the difference between pISC for recall 'twist' to movie 'spoiled' and pISC for recall 'twist' to movie 'twist'. Circled areas highlight the regions that showed significant effects in the encoding-recall analysis across subjects (**Figure 3B**), where pISC for recall 'twist' to movie 'spoiled' was greater than pISC for recall 'twist' to movie 'twist'. The area with the highest correlation among these areas (which is the highest correlation in the entire map as well) is highlighted with a black continuous-line circle. The scatter plot on the left depicts the correlation in this area. 'Critical scenes' that were used in the main analyses (**Figures 2–3**) are shown in blue on the scatter plot. The rest of the scenes are shown in red. Correlation was calculated across all 18 scenes.

**Appendix 1—table 1.** The output of condition x group ANOVA in each ROI (each row). 'Condition' columns showthe average of the univariate response in each ROI for a given condition. Statistics corresponding to eacheffect are shown in the five right columns. p Values marked as significant (*) did not pass correction formultiple comparison.

| | condition: movie | | | condition: recall | | | DFn | DFd | F | p | effect |
|---|---|---|---|---|---|---|---|---|---|---|---|
| | spoiled | twist | no-twist | spoiled | twist | no-twist | | | | | |
| LH_Default_Temp_1 | –0.02 | –0.025 | –0.038 | 0.188 | 0.106 | 0.117 | 2 | 54 | 1.237 | 0.298 | group |
| | | | | | | | 2 | 54 | 0.628 | 0.537 | group x condition |
| LH_Default_Temp_4 | –0.002 | –0.033 | –0.024 | 0.131 | 0.148 | –0.034 | 2 | 54 | 3.649 | 0.033 * | group |
| | | | | | | | 2 | 54 | 3.279 | 0.045 * | group x condition |
| LH_Default_PFC_1 | –0.005 | –0.005 | –0.024 | 0.106 | 0.067 | 0.054 | 2 | 54 | 0.434 | 0.65 | group |
| | | | | | | | 2 | 54 | 0.125 | 0.883 | group x condition |
| LH_Default_PFC_2 | –0.027 | –0.043 | –0.053 | 0.088 | 0.006 | 0.006 | 2 | 54 | 1.432 | 0.248 | group |
| | | | | | | | 2 | 54 | 0.526 | 0.594 | group x condition |
| LH_Default_PFC_3 | 0.002 | –0.005 | –0.014 | 0.088 | 0.116 | 0.055 | 2 | 54 | 0.5 | 0.609 | group |
| | | | | | | | 2 | 54 | 0.237 | 0.789 | group x condition |
| LH_Default_PFC_4 | 0.002 | –0.02 | –0.02 | 0.031 | 0.034 | –0.008 | 2 | 54 | 0.611 | 0.546 | group |
| | | | | | | | 2 | 54 | 0.281 | 0.755 | group x condition |
| LH_Default_PFC_5 | –0.011 | –0.03 | –0.033 | 0.138 | 0.09 | 0.056 | 2 | 54 | 0.97 | 0.385 | group |
| | | | | | | | 2 | 54 | 0.318 | 0.728 | group x condition |
| LH_Default_PCC_1 | 0.018 | 0.015 | 0.013 | 0.21 | 0.201 | 0.153 | 2 | 54 | 0.381 | 0.684 | group |
| | | | | | | | 2 | 54 | 0.26 | 0.771 | group x condition |
| LH_Default_PCC_2 | 0.008 | –0.017 | –0.005 | 0.178 | 0.163 | 0.023 | 2 | 54 | 2.231 | 0.117 | group |
| | | | | | | | 2 | 54 | 2.013 | 0.143 | group x condition |
| RH_Default_Par_1 | 0.037 | 0.025 | 0.013 | 0.172 | 0.252 | 0.043 | 2 | 54 | 3.151 | 0.0507~ | group |
| | | | | | | | 2 | 54 | 2.373 | 0.102 | group x condition |
| RH_Default_Temp_1 | –0.001 | –0.024 | –0.025 | 0.135 | 0.13 | 0.078 | 2 | 54 | 0.76 | 0.472 | group |
| | | | | | | | 2 | 54 | 0.249 | 0.78 | group x condition |
| RH_Default_Temp_3 | –0.014 | –0.024 | –0.026 | 0.242 | 0.239 | 0.188 | 2 | 54 | 0.374 | 0.689 | group |
| | | | | | | | 2 | 54 | 0.184 | 0.831 | group x condition |
| RH_Default_PFCm_1 | 0.006 | 0.003 | –0.005 | 0.102 | 0.126 | 0.067 | 2 | 54 | 0.437 | 0.648 | group |
| | | | | | | | 2 | 54 | 0.214 | 0.807 | group x condition |
| RH_Default_PFCm_2 | 0.009 | –0.004 | –0.014 | 0.138 | 0.12 | 0.074 | 2 | 54 | 0.693 | 0.504 | group |
| | | | | | | | 2 | 54 | 0.169 | 0.844 | group x condition |
| RH_Default_PCC_2 | 0.031 | 0.005 | 0.005 | 0.219 | 0.214 | 0.073 | 2 | 54 | 2.41 | 0.099 | group |
| | | | | | | | 2 | 54 | 1.579 | 0.215 | group x condition |

