## [Editor Report]

This study presents an important extension of recent work investigating the encoding and recall of narratives in the default mode network. The design is clever, allowing the authors to conduct multiple targeted analyses. The results are compelling and will be of interest to cognitive neuroscientists working on memory and naturalistic paradigms.

---

## [Decision Letter]

**Decision letter after peer review:**

Thank you for submitting your article "Here's the twist: How the brain updates the representations of naturalistic events as our understanding of the past changes" for consideration by *eLife*. Your article has been reviewed by 3 peer reviewers, one of whom is a member of our Board of Reviewing Editors, and the evaluation has been overseen by Floris de Lange as the Senior Editor. The following individual involved in the review of your submission has agreed to reveal their identity: Zachariah M. Reagh (Reviewer #3).

Essential revisions:

1) Better introduce the method and specific hypotheses, including the focus on the DMN.

2) Include additional analyses to provide more insight into the processes/mechanisms leading to the observed results, such as whether these reflect overall (univariate) activity differences, are scene-specific, are specific to critical (vs non-critical) scenes, regional (DMN) specificity, etc. (see suggestions by R1 and R3)

3) Present results when including all 18 scenes (note that this also allows you to address some of the concerns listed under #2 above).

4) More in-depth discussion about the possible role(s) of DMN regions in memory updating, including the distinct roles of individual regions, taking into account the partly divergent results across analyses.

*Reviewer #1 (Public Review):*

This fMRI study investigated how memories are updated after reinterpreting past events. Participants watched a movie and subsequently recalled individual scenes from that movie. Importantly, the movie ends with a twist that changes the interpretation of earlier scenes in the movie. One group of participants watched the movie with the twist at the end, one group did not get to see the twist, and a third group was already informed about this twist before watching the movie. Analyses compared the similarity of activity patterns to (encoded or recalled) events across participants within regions of the default mode network (DMN). The design allowed for multiple relevant comparisons, confirming the prediction that activity patterns in DMN regions reflect the (re)interpretation of the movie (during movie viewing and/or during recall).

The study is well-designed and executed. The inclusion of multiple analyses involving distinct comparisons strengthens the evidence for the role of the DMN in memory updating.

The following points may be relevant to consider:

1. The cross-participant pattern analysis method used here is not standard, with such analyses typically done within participants (or across participants, but after aligning representational spaces). Considering individual variability in functional organization, the method is likely only sensitive to coarse-scale patterns (e.g., anterior vs posterior parts of an ROI). This is not necessarily a weakness but is relevant when interpreting the results.

2. Unlike previous work, analyses are not testing for scene-specific information. Rather, each scene is treated separately to establish between-group differences, and results are averaged across scenes. This raises the question of whether the patterns reflect scene-specific information or generic group differences. For example, knowing the twist may increase overall engagement, both when viewing the movie (spoiled group) and when recalling it (spoiled group + twist group). The DMN may be particularly sensitive to such differences in overall engagement.

3. The study does not reveal what the DMN represents about the movie, such that its activity changes after knowing the twist. The Discussion briefly mentions that it may reflect the state of the observer, related to the belief about the identity of the doctor. This suggests a link to the theory of mind/mentalizing, but this is not made explicit. Alternatively, the DMN may be involved in the conflict (or switching) between the two interpretations.

4. The design has many naturalistic aspects, but it is also different from real life in that the critical twist involves a ghost. Furthermore, all results are based on one movie with a specific plot twist. It is thus not clear whether similar results would be obtained with other and more naturalistic plot twists.

5. Only 7 scenes (out of 18) were included in the analysis. It is not clear if/how the results depend on the selection of these 7 scenes.

*Reviewer #1 (Recommendations for the authors):*

1. The interpretation of the results critically relies on a good understanding of the method. Therefore, please introduce and explain the method more explicitly (either in the Introduction or the Results), making clear how it differs from related methods (including the temporal correlation method used in previous papers of this group) and what it may (and may not) be sensitive to. Also, please discuss the results in the context of what this method can reveal, i.e., what kind of representation can be revealed at this spatial scale.

2. This point could be addressed in additional analyses (e.g., by testing whether similar results are found when randomly pairing scenes in the analyses). Alternatively, this would be relevant to discuss and/or exclude in other ways.

3. Please include additional discussion about the putative role of the DMN in relation to the twist manipulation.

4. This could be discussed as a limitation of the study.

5. The scene selection procedure would need to be described in the Results section, following the behavioral results (as it was based on those results). It would be relevant to know how and when this was decided, particularly whether this was decided before or after analyses were conducted (and results inspected), and why 18 scenes were originally included. Also, the current scene selection appears somewhat arbitrary. For example, scene #1 was included because it showed a high twist score, but it is not clear what this twist score is or whether there were scenes with a similarly high twist score. Ideally, the authors would also present the results when including all 18 scenes (e.g., as Supplementary file).

Additional comments:

The Introduction does not sufficiently make clear why the DMN is the main focus of the study. Readers may be unfamiliar with the DMN, so it would be relevant to briefly introduce the DMN (e.g., which regions are part of it, how it is defined, what it does, etc.). This could then lead to a more focused motivation for why these regions were specifically relevant for the current study.

Considering the coarse spatial scale, results may similarly come out of simpler univariate analyses. If so, this would inform (and simplify) the interpretation of the results. For example, rather than using correlation as a measure of similarity, you could take the absolute activation difference (e.g., between Participant 1 of Group 1 and the average of Group 2), averaged across the voxels of the ROI (or at the voxel level, in a whole-brain analysis). Please include such analyses, or describe why they would not be informative.

If possible, please include the interaction analysis of Figure 2c also in the whole brain analysis (Supp figure 1). It may be easiest, for comparison, to include two Supp figures corresponding to the two main results figures (same layout, etc).

If I understand the analysis correctly, it is done separately for each of the 7 scenes and results were then averaged. To get more insight into the results, it may be informative to know which scenes showed the strongest correlation difference (e.g., within the ROIs showing an overall effect). This analysis may be most powerful when including all 18 scenes, correlating the effect of interest with the twist score from the behavioral data (i.e., correlate across scenes rather than participants, as in Supp Figure 2). This would more directly relate the results to the twist manipulation.

Please motivate how you determined the sample size.

What was the duration of the 18 scenes? How long did participants take to recall them? Did this differ across conditions?

*Reviewer #2 (Public Review):*

In this manuscript titled "Here's the twist: How the brain updates the representations of naturalistic events as our understanding of the past changes", the authors reported a study that examined how new information (manipulated as a twist at the end of a movie) changes the neural representations in the default mode network (DMN) during the recall of prior knowledge. Three groups of participants were compared - one group experienced the twist at the end, one group never experienced the twist, and one group received a spoiler at the beginning. At retrieval, participants received snippets of 18 scenes of the movie as cues and were asked to freely describe the events of each scene and to provide the most accurate interpretation of the scene, given the information they gathered throughout watching.

All three groups were highly accurate in the recall of content. The groups that experienced the twist at the end as well as at the beginning as a spoiler showed a higher twist score (the extent to which twist information was incorporated into the recall), while seemingly also keeping the interpretation without the twist ("Doctor representation") intact. Neurally, several regions in the DMN showed significant interaction effects in their neural similarity patterns (based on intersubject pattern correlation), indicating a change in interpretation between encoding and recall in the twist group uniquely, presumably reflecting memory updating.

Several points that I think should be addressed to strengthen the manuscript:

1) The results from encoding-retrieval similarity analysis (particularly the one depicted in Figure 3B) don't match the results from encoding/retrieval interaction (particularly those shown in Figure 2C). While they were certainly based on different comparisons, I would think that both analyses were set up to test for memory updating. Can the authors comment on this divergence in results?

2) The recall task was self-paced. Can reaction time information be provided on how long participants needed to recall? Did this differ across groups? Presumably in the twist group and spoiled group participants might have needed a longer time to incorporate both the original and twist interpretation. How was the length difference across events taken into consideration in the beta estimates? Also, is there an order effect, such that one type of interpretation tended to be recalled first?

3) The correlation analysis between neural pattern change and behavioral twist score is based on a small sample size and does not seem to be well suited to test the postulation of the authors, namely that some participants may hold both interpretations in their memory. Interestingly, the twist score of the spoiled group was similar to the twist group, indicating participants in this group might have held both interpretations as well. Could this observation be leveraged, for example by combining both groups (hence better powered with larger sample size), in order to relate individual differences in neural similarity patterns and behavioral tendency to hold both interpretations?

4) Several regions within the DMN were significant across the analysis steps, specifically the angular gyrus, middle temporal cortex, and medial PFC. Can the authors provide more insights on how these widely distributed regions may act together to enable memory updating? The discussion on the main findings is largely at a rather superficial level about DMN, or focuses specifically on vmPFC, but neglects the distributed regions that presumably function interactively.

*Reviewer #2 (Recommendations for the authors):*

The prediction legends in Figures 2 and 3 are very helpful to follow the contrasts involved and the predictions made. My only suggestion is to increase the word font, as in the current version it is somewhat difficult to read.

*Reviewer #3 (Public Review):*

Zadbood and colleagues investigated the way key information used to update interpretations of events alter patterns of activity in the brain. This was cleverly done by the use of "The Sixth Sense," a film featuring a famous "twist ending," which fundamentally alters the way the events in the film are understood. Participants were assigned to three groups: (1) a Spoiled group, in which the twist was revealed at the outset, (2) a Twist group, who experienced the film as normal, and (3) a No-Twist group, in which the twist was removed. Participants were scanned while watching the movie and while performing cued recall of specific scenes. Verbal recall was scored based on recall success, and evidence for descriptive bias toward two ways of understanding the events (specifically, whether a particular character was or was not a ghost). Importantly, this allowed the authors to show that the Twist group updated their interpretation. The authors focused on regions of the Default Mode Network (DMN) based on prior studies showing responsiveness to naturalistic memory paradigms in these areas and analyzed the fMRI data using intersubject pattern similarity analysis. Regions of the DMN carried patterns indicative of story interpretation. That is, encoding similarity was greater between the Twist and No-Twist groups than in the Spoiled group, and retrieval similarity was greater between the Twist and Spoiled groups than in the No-Twist group. The Spoiled group also showed greater pattern similarity with the Twist group's recall than the No-Twist group's recall. The authors also report a weaker effect of greater pattern similarity between the Spoiled group's encoding and the Twist group's recall than between the Twist group's own encoding and recall. Together, the data all converge on the point that one's interpretation of an event is an important determinant of the way it is represented in the brain.

This is a really nice experiment, with straightforward predictions and analyses that support the claims being made. The results build directly on a prior study by this research group showing how interpretational differences in a narrative drive distinct neural representations (Yeshurun et al., 2017), but extend an understanding of how these interpretational differences might work retrospectively. I do not have any serious concerns or problems with the manuscript, the data, or the analyses. However I have a few points to raise that, if addressed, would make for a stronger paper in my opinion.

1) My most substantive comment is that I did not find the interpretive framework to be very clear with respect to the brain regions involved. The basic effects the authors report strongly support their claims, but the particular contributions to the field might be stronger if the interpretations could be made more strongly or more specifically. In other words: the DMN is involved in updating interpretations, but how should we now think about the role of the DMN and its constituent regions as a result of this study? There are a number of ideas briefly presented about what the DMN might be doing, but it just did not feel very coherent at times. I will break this down into a few more specific points:

While many of us would agree that the DMN is likely to be involved in the phenomena at hand, I did not find that the paper communicated the logic for singularly focusing on this subset of regions very compellingly. The authors note a few studies whose main results are found in DMN regions, but I think that this could stand to be unpacked in a more theoretically interesting way in the Introduction.

Relatedly, I found the summary/description of regional effects in the Discussion to be a bit unsatisfying. The various pattern similarity comparisons yielded results that were actually quite nonoverlapping among DMN regions, which was not really unpacked. To be clear, it is not a 'problem' that the regional effects varied from comparison to comparison, but I do think that a more theoretical exploration of what this could mean would strengthen the paper. To the authors' credit, they describe mPFC effects through the lens of schemas, but this stands in contrast to many other regions which do not receive much consideration.

Finally, although there is evidence that regions of the DMN act in a coordinated way under some circumstances, there is also ample evidence for distinct regional contributions to cognitive processes, memory being just one of them (e.g., Cooper & Ritchey, 2020; Robin & Moscovitch, 2017; Ranganath & Ritchey, 2012). The authors themselves introduce the idea of temporal receptive windows in a cortical hierarchy, and while DMN regions do appear to show slower temporal drift than sensory areas, those studies show regional differences in pattern stability across time even within DMN regions. Simply put, it is worth considering whether it is ideal to treat the DMN as a singular unit.

2) I think that some direct comparison to regions outside the DMN would speak to whether the DMN is truly unique in carrying the key representations being discussed here. I was reluctant to suggest this because I think that the authors are justified in expecting that DMN regions would show the effects in question. However, there really is no "null" comparison here wherein a set of regions not expected to show these effects (e.g., a somatosensory network, or the frontoparietal network) in fact do not show them. There are not really controls or key differences being hypothesized across different conditions or regions. Rather, we have a set of regions that may or may not show pattern similarity differences to varying degrees, which feels very exploratory. The inclusion of some principled control comparisons, etc. would bolster these findings. The authors do include a whole-brain analysis in Supplementary Figure 1, which indeed produced many DMN regions. However, notably, regions outside the DMN such as the primary visual cortex and mid-cingulate cortex appear to show significant effects (which, based on the color bar, might actually be stronger than effects seen in the DMN). Given the specificity of the language in the paper in terms of the DMN, I think that some direct regional or network-level comparison is needed.

3) If I understand correctly, the main analyses of the fMRI data were limited to across-group comparisons of "critical scenes" that were maximally affected by the twist at the end of the movie. In other words, the analyses focused on the scenes whose interpretation hinged on the "doctor" versus "ghost" interpretation. I would be interested in seeing a comparison of "critical" scenes directly against scenes where the interpretation did not change with the twist. This "critical" versus "non-critical" contrast would be a strong confirmatory analysis that could further bolster the authors' claims, but on the other hand, it would be interesting to know whether the overall story interpretation led to any differences in neural patterns assigned to scenes that would not be expected to depend on differences in interpretation. (As a final note, such a comparison might provide additional analytical leverage for exploring the effect described in Figure 3B, which did not survive correction for multiple comparisons.)

4) I appreciate the code being made available and that the neuroimaging data will be made available soon. I would also appreciate it if the authors made the movie stimulus and behavioral data available. The movie stimulus itself is of interest because it was edited down, and it would be nice for readers to be able to see which scenes were included.

To sum up, I think that this is a great experiment with a lot of strengths. The design is fairly clean (especially for a movie stimulus), the analyses are well reasoned, and the data are clear. The only weaknesses I would suggest addressing are with regards to how the DMN is being described and evaluated, and the communication of how this work informs the field on a theoretical level.

*Reviewer #3 (Recommendations for the authors):*

I want to emphasize that I am a big fan of the study and the approach overall. It is very well done, the results are clear and interesting, and the paper is overall well constructed. Below, I will expand on some of the points I raised in the public review, and provide some suggestions for how they might be addressed.

1) My first point dealt with asking for a somewhat stronger explanation for a focus on the DMN, and the question of how the reader should update their model of the DMN on the basis of these results. The updating phenomenon is certainly interesting, and as I noted in the public review, a focus on the DMN is sensible. However, I think this focus could be more clearly justified. The results themselves were a bit inconsistent in terms of which specific DMN regions showed pattern similarity effects, and I found myself wondering what this might mean. A bit more unpacking of this could be helpful. Beyond these points, however, there are theoretical frameworks arguing pretty compellingly that subsystems of the DMN may uniquely contribute to cognition (e.g., Maureen Ritchey's and Morris Moscovitch's ideas). This is a bit at odds with treating the DMN as a single unit. While I want to be clear that I do not think it is necessarily wrong to do so in this case, it does warrant some consideration.

2) As I noted in the public review, I hesitated to bring this up. However, some direct comparison of DMN vs. non-DMN would really bolster the results and the claims in my opinion. This would also go a long way in addressing my points above.

3) I was not sure when writing this comment whether this "critical" versus "non-critical" analysis was tried, but in my view, it could serve to bolster the findings being presented in the paper currently. And as I noted in the public review, on the other hand, it would be interesting to know if the effects of the twist on neural patterns reached beyond the "critical" scenes into a general interpretation of the story.

---

## [Author Response]

Essential revisions:1) Better introduce the method and specific hypotheses, including the focus on the DMN.

We addressed this by adding the following paragraphs to Introduction and the beginning of the Results section.

Paragraph added to Introduction:

“The brain’s default mode network (DMN)—comprising the posterior medial cortex, medial prefrontal cortex, temporoparietal junction, and parts of anterior temporal cortex—was originally described as an intrinsic or “task-negative” network, activated when participants are not engaged with external stimuli (Raichle et al. 2001, Buckner et al. 2008). […] Building on this foundation of prior work on the DMN, we asked whether we could find neural evidence for the retroactive influence of new knowledge on past memories.”

Paragraph added to the beginning of Results section:

“We used intersubject pattern similarity analysis (intersubject pattern correlation: pISC, see Methods) to compare the neural event representations between groups. […] We focused our analyses on a predetermined selection of movie scenes (i.e., 7 “critical scenes” out of 18 total scenes) in which the *Doctor* or *Ghost* interpretation of the main character in the movie would dramatically change the overall interpretation of those scenes. Selection of these scenes was based on ratings from four raters asked to quantify the influence of the twist on the interpretation of each scene (see Methods).”

2) Include additional analyses to provide more insight into the processes/mechanisms leading to the observed results, such as whether these reflect overall (univariate) activity differences, are scene-specific, are specific to critical (vs non-critical) scenes, regional (DMN) specificity, etc. (see suggestions by R1 and R3)

Thank you for summarizing the reviewers’ suggestions. We added new analyses for each of these suggestions.

A univariate analysis was performed and reported in the Results section (end of Neural representation of the twist information during cued recall):

“To test whether our reported results were mainly driven by the similarities and differences in multivariate spatial patterns of neural representations, as opposed to by univariate regional-average response magnitudes, we ran a univariate analysis in each ROI. This analysis revealed no significant effect of group (“spoiled”, “twist”, “no-twist”) or interaction between group and condition (movie, recall) (Table 1, see Methods for details).”

Scene-related analyses (critical vs. non-critical scenes analysis and scene specificity analysis) are reported in a new Results section entitled “The role of scene content”:

“The role of scene content

In the prior analyses, we focused on “critical scenes”, selected based on ratings from four raters who quantified the influence of the twist on the interpretation of each scene (see Methods). […] However, they appear to match the direction of our main analyses; with greater statistical power, analyses of this sort may provide insights into how neural event representations are updated in a scene-specific manner.”

DMN specificity is further evaluated in a new Results section entitled “The changes in neural representations beyond DMN”:

“Changes in neural representations beyond the DMN

We focused our core analyses on regions of the default mode network. Prior work has shown that multimodal neural representations of naturalistic events (e.g. movie scenes) are similar across encoding (movie-watching or story-listening) and verbal recall of the same events in the DMN (Chen et al., 2017; Zadbood et al., 2017). […] In the encoding-encoding comparison, several ROIs from the visual and somatomotor networks showed relatively strong effects as well (see Discussion).”

The whole brain analysis figure was modified to add the interaction analysis (asked by reviewer 1) and adding text for further discussion of anterior cingulate cortex and dorsolateral prefrontal cortex regions:

“In addition, we qualitatively reproduced our results by performing an ROI-based whole brain analysis (Appendix-Figure 3, p < 0.01 uncorrected). This analysis confirmed the importance of DMN regions for updating neural event representations. However, strong differences in pISC in the hypothesized direction were also observed in a handful of other non-DMN regions, including ROIs partly overlapping with anterior cingulate cortex and dorsolateral prefrontal cortex (see Discussion).”

Sections added to the Methods corresponding to the new analyses:

“The same pISC procedures were performed in the regions of the visual and somatosensory cortices (Appendix-Figure 2), as well as across the whole brain (Appendix-Figure 3). All regions were extracted from the 100-parcel Shaeffer where each region is assigned to one of seven resting-state networks (Shaeffer et al. 2018). […] To better understand the role of “critical scenes” in this relationship, those scenes were marked as blue on the scatter plots (Appendix-Figure 4, left panels).”

3) Present results when including all 18 scenes (note that this also allows you to address some of the concerns listed under #2 above).

This analysis was added in the “The role of scene content” section and Figure 4-A, described above.

4) More in-depth discussion about the possible role(s) of DMN regions in memory updating, including the distinct roles of individual regions, taking into account the partly divergent results across analyses.

The following paragraphs were added to the Discussion to address these points:

“In addition to mPFC, right precuneus and parts of temporal cortex exhibited significantly higher pattern similarity in the “twist” and “spoiled” groups who recalled the movie with the same interpretation. […] Future extensions of this work may benefit from using functional alignment methods (Haxby et al. 2020, Chen et al. 2015) to capture more fine-grained event representations which are shared across participants.”

Reviewer #1 (Public Review):This fMRI study investigated how memories are updated after reinterpreting past events. Participants watched a movie and subsequently recalled individual scenes from that movie. Importantly, the movie ends with a twist that changes the interpretation of earlier scenes in the movie. One group of participants watched the movie with the twist at the end, one group did not get to see the twist, and a third group was already informed about this twist before watching the movie. Analyses compared the similarity of activity patterns to (encoded or recalled) events across participants within regions of the default mode network (DMN). The design allowed for multiple relevant comparisons, confirming the prediction that activity patterns in DMN regions reflect the (re)interpretation of the movie (during movie viewing and/or during recall).The study is well-designed and executed. The inclusion of multiple analyses involving distinct comparisons strengthens the evidence for the role of the DMN in memory updating.The following points may be relevant to consider:1. The cross-participant pattern analysis method used here is not standard, with such analyses typically done within participants (or across participants, but after aligning representational spaces). Considering individual variability in functional organization, the method is likely only sensitive to coarse-scale patterns (e.g., anterior vs posterior parts of an ROI). This is not necessarily a weakness but is relevant when interpreting the results.

We agree with the reviewer that functional misalignment might have played against us here. We designed this study as a natural successor of our previous work in which we captured reliable and multimodal scene-specific cross-participant pattern similarity during encoding and recall in standard space. In this revised version, we provide further evidence on how scene content is captured and influences our results. Nonetheless, we agree with your comment and add the following section to the discussion to encourage considering this point while interpreting the results.

“Moreover, our current method relies on averaging spatially-coarse activity patterns across subjects (and time points within an event). Future extensions of this work may benefit from using functional alignment methods (Haxby et al 2020, Chen et al 2015) to capture more fine-grained event representations which are shared across participants.”

2. Unlike previous work, analyses are not testing for scene-specific information. Rather, each scene is treated separately to establish between-group differences, and results are averaged across scenes. This raises the question of whether the patterns reflect scene-specific information or generic group differences. For example, knowing the twist may increase overall engagement, both when viewing the movie (spoiled group) and when recalling it (spoiled group + twist group). The DMN may be particularly sensitive to such differences in overall engagement.

You have brought up great points. We addressed them in two ways: (1) We ran a univariate analysis in each DMN ROI to look at the role of overall regional-average response magnitude in our results. We did not observe a significant effect of group or an interaction between group and condition. (2) We ran a scene-specificity analysis in a new Results section entitled “The role of scene content” (Figure 4). This section is focused on comparing interaction index (Figure 2C), as an indicator of memory updating, under different manipulations. Interaction index reflects the reversal of neural similarity during encoding and recall. Our results suggest that we don’t see the same effects if we shuffle the scene labels and recompute the pattern similarity analyses. Please see added text below:

“To test whether our reported results were mainly driven by the similarities and differences in multivariate spatial patterns of neural representations, as opposed to by univariate regional-average response magnitudes, we ran a univariate analysis in each ROI. This analysis revealed no significant effect of group (“spoiled”, “twist”, “no-twist”) or interaction between group and condition (movie, recall) (Table 1, see Methods for details) […] Next, to determine whether scene-specific neural event representations—as opposed to coarser differences in general mental state across all scenes with similar interpretations—drive our observed pISC differences, we shuffled the labels of critical scenes within each group before calculating and comparing pISC across groups. By repeating this procedure 1000 times and recalculating the interaction index at each iteration, we constructed a null distribution of interaction indices for shuffled critical scenes (light magenta distributions in Figure 4B). In 12 out of 24 DMN regions, interaction indices were statistically significant based on the shuffled-scene distribution (p < .025, FDR controlled at q < .05). All of these 12 regions were among the ROIs that showed meaningful effects in our original analysis (Figure 2C). Regions with significant scene-specific interaction effects are marked as blue dots with black borders in Figure 4B. Overall, the findings from this analysis confirm that our results are driven by changes to scene-specific representations.”

3. The study does not reveal what the DMN represents about the movie, such that its activity changes after knowing the twist. The Discussion briefly mentions that it may reflect the state of the observer, related to the belief about the identity of the doctor. This suggests a link to the theory of mind/mentalizing, but this is not made explicit. Alternatively, the DMN may be involved in the conflict (or switching) between the two interpretations.

Great points. We added to the discussion about the role of mentalizing network and in the particular temporo-parietal cortex. About your last point, we think our whole brain findings outside DMN (ACC and dlPFC) might relate to that point. We discussed these further in the paper.

“We performed two targeted analyses to look for evidence of memory updating across encoding and recall: the interaction analysis (Figure 2C) and the encoding-recall analysis (Figure 3). […] These results suggest that both the "spoiled" manipulation and the "twist" may recruit top-down control and conflict monitoring processes during naturalistic viewing and recall.

4. The design has many naturalistic aspects, but it is also different from real life in that the critical twist involves a ghost. Furthermore, all results are based on one movie with a specific plot twist. It is thus not clear whether similar results would be obtained with other and more naturalistic plot twists.

We added this as a limitation of the study.

“Our findings provide further insight into the functional role of the DMN. However, these results have been obtained using only one movie. While naturalistic paradigms better capture the complexity of real life and provide greater ecological generalizability than highly-controlled experimental stimuli and tasks (Nastase et al., 2020), they are still limited by the properties of the particular naturalistic stimulus used. For example, this movie—including the twist itself—hinges on suspension of disbelief about the existence of ghosts. Future work is needed to extend our findings about updating event memories to a broader class of naturalistic stimuli: for example, movies with different kinds of (non-supernatural) plot twists, spoken stories with twist endings, or using autobiographical real-life situations where new information (e.g. discovering a longtime friend has lied about something important) triggers re-evaluation of the past (e.g. reinterpreting their friend’s previous actions).”

5. Only 7 scenes (out of 18) were included in the analysis. It is not clear if/how the results depend on the selection of these 7 scenes.

Thank you for bringing this up. These scenes were pre-selected for the analyses, as they are the only scenes that are rated high by our independent raters (not study participants) on “twist influence”, meaning that knowing the twist could dramatically change their interpretation. So, we had a priori reasons to hypothesize that the effect will be strong in these scenes. To address your point, we report results by including all 18 scenes in a new Results section entitled “The role of scene content” and in Figure 4A. While the effect was weaker for all scenes it was still apparent in this conservative analysis. As expected, however, including 7 critical scenes produces stronger results than including all scenes or the uncritical scenes (all minus critical scenes). Please see the “The role of scene content” in Results and in Figure 4 for more detailed information.

Reviewer #1 (Recommendations for the authors):1. The interpretation of the results critically relies on a good understanding of the method. Therefore, please introduce and explain the method more explicitly (either in the Introduction or the Results), making clear how it differs from related methods (including the temporal correlation method used in previous papers of this group) and what it may (and may not) be sensitive to. Also, please discuss the results in the context of what this method can reveal, i.e., what kind of representation can be revealed at this spatial scale.

Thank you for bringing this up. We added text to the beginning of results to address this:

“We used intersubject pattern similarity analysis (intersubject pattern correlation: pISC, see Methods) to compare the neural event representations between groups. […] Selection of these scenes was based on ratings from four raters asked to quantify the influence of the twist on the interpretation of each scene (see Methods).”

2. This point could be addressed in additional analyses (e.g., by testing whether similar results are found when randomly pairing scenes in the analyses). Alternatively, this would be relevant to discuss and/or exclude in other ways.

Thank you. We ran an additional analysis for this as stated in our response in the public section.

3. Please include additional discussion about the putative role of the DMN in relation to the twist manipulation.

We added these paragraphs to the discussion.

“In addition to mPFC, right precuneus and parts of temporal cortex exhibited significantly higher pattern similarity in the “twist” and “spoiled” groups who recalled the movie with the same interpretation. […] This transformation could affect how different regions and sub-systems of DMN represent memories, and suggests that the concerted activity of multiple subsystems and neural mechanisms might be at play during encoding, recall and successful updating of naturalistic event memories.”

4. This could be discussed as a limitation of the study.

We discussed this as a limitation of the study as you suggested.

“Our findings provide further insight into the functional role of the DMN. However, these results have been obtained using only one movie. While naturalistic paradigms better capture the complexity of real life and provide greater ecological generalizability than highly-controlled experimental stimuli and tasks (Nastase et al., 2020), they are still limited by the properties of the particular naturalistic stimulus used. For example, this movie—including the twist itself—hinges on suspension of disbelief about the existence of ghosts. Future work is needed to extend our findings about updating event memories to a broader class of naturalistic stimuli: for example, movies with different kinds of (non-supernatural) plot twists, spoken stories with twist endings, or using autobiographical real-life situations where new information (e.g. discovering a longtime friend has lied about something important) triggers re-evaluation of the past (e.g. reinterpreting their friend’s previous actions).”

5. The scene selection procedure would need to be described in the Results section, following the behavioral results (as it was based on those results). It would be relevant to know how and when this was decided, particularly whether this was decided before or after analyses were conducted (and results inspected), and why 18 scenes were originally included. Also, the current scene selection appears somewhat arbitrary. For example, scene #1 was included because it showed a high twist score, but it is not clear what this twist score is or whether there were scenes with a similarly high twist score. Ideally, the authors would also present the results when including all 18 scenes (e.g., as Supplementary file).

Thank you for your comment. We tried to make the scene selection procedure more clear at the end of the Results section opening and also the beginning of the new Results section (The role of scene content). We also made some edits in the related Methods section. To your points, our scene selection procedure was separate from subjects’ behavioral twist score. But interestingly, it did match it. Before running the study and based on our internal rating of the content of the scenes (by AZ from the authors), we knew those selected scenes were perhaps the most critical ones for our questions. Ratings from three independent raters confirmed this selection. However, as the doctor was present in most of the other scenes of the movie and we were not sure how the twist would be reflected in subjects’ behavioral recall of unrelated scenes, we included all 18 scenes of the movie in the study. Behavioral recall “twist scores” confirmed that those originally hypothesized scenes contained the highest behavioral reflection of the twist information. We made this procedure more clear by adding text to the beginning of the new Results section (The role of scene content) and the Methods section (Timestamping and scene selection). About scene 1, its twist score is the highest after the main 6 scenes and equals to 1 other scene. As stated in the text, we included this scene as it was the first time subjects saw the doctor after watching the twist and needed to retrieve him in this new context (*Ghost* instead of *Doctor*). So, we hypothesized it might be an important scene to keep and subjects’ twist scores confirmed it.

To summarize, we had pre-selected a set of scenes that we thought would be particularly relevant before we ran the study. But we also had the post-encoding twist scores from participants’ recall in hand (which perfectly matched our prior movie-based selection) at the time of analysis. Results with all scenes are also presented in the revision of the paper. The following snippets show the related edits in the paper:

“We focused our analyses on a predetermined selection of movie scenes (i.e., 7 “critical scenes” out of 18 total scenes) in which the *Doctor* or *Ghost* interpretation of the main character in the movie would dramatically change the overall interpretation of those scenes. Selection of these scenes was based on ratings from four raters asked to quantify the influence of the twist on the interpretation of each scene (see Methods).”

“In the prior analyses, we focused on “critical scenes”, selected based on ratings from four raters who quantified the influence of the twist on the interpretation of each scene (see Methods). An independent post-experiment analysis of the verbal recall behavior of the fMRI participants yielded “twist scores” that were also highest for these scenes; that is, the expected and perceived effect of twist information on recall behavior were found to match. Next, we asked whether the neural event representations reflect these differences in the twist-related content of the scenes. In other words, are the “critical scenes” with highly twist-dependent interpretations truly *critical* for our observed effects?”

“Edited method:

Timestamping and scene selection

The movie was time-stamped by an independent rater naive to the purpose and design of the experiment to identify the main scenes of the movie. All of the movie scenes with clear scene boundaries (N = 18) were selected to be used in the cued-recall task. […] Therefore, we added this scene as a seventh *critical scene* to be used in the main neural analyses.”

Additional comments:The Introduction does not sufficiently make clear why the DMN is the main focus of the study. Readers may be unfamiliar with the DMN, so it would be relevant to briefly introduce the DMN (e.g., which regions are part of it, how it is defined, what it does, etc.). This could then lead to a more focused motivation for why these regions were specifically relevant for the current study.

This paragraph was added to the introduction:

“The brain’s default mode network (DMN)—comprising the posterior medial cortex, medial prefrontal cortex, temporoparietal junction, and parts of anterior temporal cortex—was originally described as an intrinsic or “task-negative” network, activated when participants are not engaged with external stimuli (Raichle et al. 2001, Buckner et al. 2008). […] Building on this foundation of prior work on the DMN, we asked whether we could find neural evidence for the retroactive influence of new knowledge on past memories.”

Considering the coarse spatial scale, results may similarly come out of simpler univariate analyses. If so, this would inform (and simplify) the interpretation of the results. For example, rather than using correlation as a measure of similarity, you could take the absolute activation difference (e.g., between Participant 1 of Group 1 and the average of Group 2), averaged across the voxels of the ROI (or at the voxel level, in a whole-brain analysis). Please include such analyses, or describe why they would not be informative.

We included a univariate analysis by looking at differences in absolute activation values across groups and conditions in each ROI. Running an ANOVA in each ROI revealed no significant effect of group or interaction between group and condition. We found a significant effect of group in two ROIs that didn’t pass correction for multiple comparisons. As you suggested, univariate activation values in some ROIs were more similar in conditions that were associated with the same interpretation (average values are reported in Table 1). But based on our results here and also the results of other analyses in the manuscript (in particular, our new “scene-specificity” analysis, where we showed that our results go away when we permute scene labels within group, without altering mean univariate activation), group differences in mean univariate activation alone do not seem to explain our results.

If possible, please include the interaction analysis of Figure 2c also in the whole brain analysis (Supp figure 1). It may be easiest, for comparison, to include two Supp figures corresponding to the two main results figures (same layout, etc).

Thanks for pointing this out. It was added to Appendix-Figure 2.

If I understand the analysis correctly, it is done separately for each of the 7 scenes and results were then averaged. To get more insight into the results, it may be informative to know which scenes showed the strongest correlation difference (e.g., within the ROIs showing an overall effect). This analysis may be most powerful when including all 18 scenes, correlating the effect of interest with the twist score from the behavioral data (i.e., correlate across scenes rather than participants, as in Supp Figure 2). This would more directly relate the results to the twist manipulation.

Thank you for this interesting suggestion. We explored this further and you can find the results at the end of the Results section on the role of scene content and in Appendix-Figure 4. We did not see strong results but the direction of findings were interesting and complement our main analysis so we included this as a supplemental analysis/appendix figure.

“To further evaluate the relationship between scene-specific twist information in the brain and behavior, we ran an exploratory analysis which was focused on the changes in the neural event representations during recall of the “twist” group and their corresponding recall behavior. […] However, they appear to match the direction of our main analyses; with greater statistical power, analyses of this sort may provide insights into how neural event representations are updated in a scene-specific manner.”

Please motivate how you determined the sample size.

Our sample size was decided based on our previous work in which we captured scene-specific pattern similarity across encoding, recall, and listening (18 subjects per group in Zadbood et al. 2017, 17 subjects in Chen et al. 2016) and differences in brain response while listening to the same story with different perspectives (20 subjects per group in Yeshurun et al. 2017). We added this information to the Methods section now. We originally collected 21 participants in each group but we had to remove participants for head motions and other reasons mentioned in the Methods. So, we ended up with 18, 19, and 20 participants per group.

What was the duration of the 18 scenes? How long did participants take to recall them? Did this differ across conditions?

We added this information to the methods section (end of *Behavioral analysis*). No significant difference between overall recall time or average recall time per scene was observed across groups. Section added to Methods:

“The average length of scenes in the 55 minute movie was 2 minutes and 10 seconds (sd = 1:59, median = 1:56, min = 00:26, max = 5:56). For the recalls, in the “spoiled” group the average recall time per scene was 39.4 seconds (sd = 13.2 s, min = 14 s, max = 67 s) for a total average of 713 seconds of recall time. In the “twist” group, the average recall time per scene was 38.5 seconds (sd = 13 s, min= 17 s, max = 69 s) for a total average of 698 seconds of recall time. In the “no-twist” group, the average recall time per scene was 37.75 seconds (sd = 19.8 s, min = 8 s, max = 73 s) for a total average of 642 seconds of recall time. No significant differences were observed between average recall time per scene or overall recall time across any two groups (according to t-tests).”

Reviewer #2 (Public Review):In this manuscript titled "Here's the twist: How the brain updates the representations of naturalistic events as our understanding of the past changes", the authors reported a study that examined how new information (manipulated as a twist at the end of a movie) changes the neural representations in the default mode network (DMN) during the recall of prior knowledge. Three groups of participants were compared - one group experienced the twist at the end, one group never experienced the twist, and one group received a spoiler at the beginning. At retrieval, participants received snippets of 18 scenes of the movie as cues and were asked to freely describe the events of each scene and to provide the most accurate interpretation of the scene, given the information they gathered throughout watching.All three groups were highly accurate in the recall of content. The groups that experienced the twist at the end as well as at the beginning as a spoiler showed a higher twist score (the extent to which twist information was incorporated into the recall), while seemingly also keeping the interpretation without the twist ("Doctor representation") intact. Neurally, several regions in the DMN showed significant interaction effects in their neural similarity patterns (based on intersubject pattern correlation), indicating a change in interpretation between encoding and recall in the twist group uniquely, presumably reflecting memory updating.Several points that I think should be addressed to strengthen the manuscript:1) The results from encoding-retrieval similarity analysis (particularly the one depicted in Figure 3B) don't match the results from encoding/retrieval interaction (particularly those shown in Figure 2C). While they were certainly based on different comparisons, I would think that both analyses were set up to test for memory updating. Can the authors comment on this divergence in results?

Thank you for your comment. Except for one ROI, the other two regions in Figure 2C are present in the interaction analysis. The ROI at the frontal pole might be hard to see from this angle but in fact it holds a high effect size in interaction analysis. So we do not see a big divergence between these two results. But taking into account the recall-recall results, we agree that there seems to be inhomogeneity. We discussed these further in the discussion.

“We performed two targeted analyses to look for evidence of memory updating across encoding and recall: the interaction analysis (Figure 2C) and the encoding-recall analysis (Figure 3). […] This transformation could affect how different regions and sub-systems of DMN represent memories, and suggests that the concerted activity of multiple subsystems and neural mechanisms might be at play during encoding, recall and successful updating of naturalistic event memories.”

2) The recall task was self-paced. Can reaction time information be provided on how long participants needed to recall? Did this differ across groups? Presumably in the twist group and spoiled group participants might have needed a longer time to incorporate both the original and twist interpretation.

This is an interesting idea. Unfortunately, we could not measure this accurately because our recall cues were snippets from the beginning of each scene with different length (selected based on content). And updating could begin from the beginning of those snippets (but we wouldn’t know when). We will consider this point in the future related designs.

How was the length difference across events taken into consideration in the beta estimates?

They were used as event durations in the GLM model.

Also, is there an order effect, such that one type of interpretation tended to be recalled first?

This is indeed hard to measure as you mentioned. We will provide the transcripts when sharing the data and hopefully this will facilitate future text-analysis work on this dataset to answer interesting questions like this.

3) The correlation analysis between neural pattern change and behavioral twist score is based on a small sample size and does not seem to be well suited to test the postulation of the authors, namely that some participants may hold both interpretations in their memory. Interestingly, the twist score of the spoiled group was similar to the twist group, indicating participants in this group might have held both interpretations as well. Could this observation be leveraged, for example by combining both groups (hence better powered with larger sample size), in order to relate individual differences in neural similarity patterns and behavioral tendency to hold both interpretations?

Even though both groups showed signs of holding both interpretations in mind, the process happening in their brain during the recall is different. In particular, we do not expect to see any updating effect in the spoiled group. So it wouldn’t seem accurate to combine these groups to test the effect of incomplete updating.

4) Several regions within the DMN were significant across the analysis steps, specifically the angular gyrus, middle temporal cortex, and medial PFC. Can the authors provide more insights on how these widely distributed regions may act together to enable memory updating? The discussion on the main findings is largely at a rather superficial level about DMN, or focuses specifically on vmPFC, but neglects the distributed regions that presumably function interactively.

Thanks for bringing this up. We added text to discussion to respond to this very valid point. Please see the added text in our response to your first point. One more snippet added to the discussion about this:

“In addition to mPFC, right precuneus and parts of temporal cortex exhibited significantly higher pattern similarity in the “twist” and “spoiled” groups who recalled the movie with the same interpretation. […] This transformation could affect how different regions and sub-systems of DMN represent memories, and suggests that the concerted activity of multiple subsystems and neural mechanisms might be at play during encoding, recall and successful updating of naturalistic event memories.”

Reviewer #2 (Recommendations for the authors):The prediction legends in Figures 2 and 3 are very helpful to follow the contrasts involved and the predictions made. My only suggestion is to increase the word font, as in the current version it is somewhat difficult to read.

We did it. Thank you for bringing it up.

Reviewer #3 (Public Review):Zadbood and colleagues investigated the way key information used to update interpretations of events alter patterns of activity in the brain. This was cleverly done by the use of "The Sixth Sense," a film featuring a famous "twist ending," which fundamentally alters the way the events in the film are understood. Participants were assigned to three groups: (1) a Spoiled group, in which the twist was revealed at the outset, (2) a Twist group, who experienced the film as normal, and (3) a No-Twist group, in which the twist was removed. Participants were scanned while watching the movie and while performing cued recall of specific scenes. Verbal recall was scored based on recall success, and evidence for descriptive bias toward two ways of understanding the events (specifically, whether a particular character was or was not a ghost). Importantly, this allowed the authors to show that the Twist group updated their interpretation. The authors focused on regions of the Default Mode Network (DMN) based on prior studies showing responsiveness to naturalistic memory paradigms in these areas and analyzed the fMRI data using intersubject pattern similarity analysis. Regions of the DMN carried patterns indicative of story interpretation. That is, encoding similarity was greater between the Twist and No-Twist groups than in the Spoiled group, and retrieval similarity was greater between the Twist and Spoiled groups than in the No-Twist group. The Spoiled group also showed greater pattern similarity with the Twist group's recall than the No-Twist group's recall. The authors also report a weaker effect of greater pattern similarity between the Spoiled group's encoding and the Twist group's recall than between the Twist group's own encoding and recall. Together, the data all converge on the point that one's interpretation of an event is an important determinant of the way it is represented in the brain.This is a really nice experiment, with straightforward predictions and analyses that support the claims being made. The results build directly on a prior study by this research group showing how interpretational differences in a narrative drive distinct neural representations (Yeshurun et al., 2017), but extend an understanding of how these interpretational differences might work retrospectively. I do not have any serious concerns or problems with the manuscript, the data, or the analyses. However I have a few points to raise that, if addressed, would make for a stronger paper in my opinion.1) My most substantive comment is that I did not find the interpretive framework to be very clear with respect to the brain regions involved. The basic effects the authors report strongly support their claims, but the particular contributions to the field might be stronger if the interpretations could be made more strongly or more specifically. In other words: the DMN is involved in updating interpretations, but how should we now think about the role of the DMN and its constituent regions as a result of this study? There are a number of ideas briefly presented about what the DMN might be doing, but it just did not feel very coherent at times. I will break this down into a few more specific points:While many of us would agree that the DMN is likely to be involved in the phenomena at hand, I did not find that the paper communicated the logic for singularly focusing on this subset of regions very compellingly. The authors note a few studies whose main results are found in DMN regions, but I think that this could stand to be unpacked in a more theoretically interesting way in the Introduction.Relatedly, I found the summary/description of regional effects in the Discussion to be a bit unsatisfying. The various pattern similarity comparisons yielded results that were actually quite nonoverlapping among DMN regions, which was not really unpacked. To be clear, it is not a 'problem' that the regional effects varied from comparison to comparison, but I do think that a more theoretical exploration of what this could mean would strengthen the paper. To the authors' credit, they describe mPFC effects through the lens of schemas, but this stands in contrast to many other regions which do not receive much consideration.Finally, although there is evidence that regions of the DMN act in a coordinated way under some circumstances, there is also ample evidence for distinct regional contributions to cognitive processes, memory being just one of them (e.g., Cooper & Ritchey, 2020; Robin & Moscovitch, 2017; Ranganath & Ritchey, 2012). The authors themselves introduce the idea of temporal receptive windows in a cortical hierarchy, and while DMN regions do appear to show slower temporal drift than sensory areas, those studies show regional differences in pattern stability across time even within DMN regions. Simply put, it is worth considering whether it is ideal to treat the DMN as a singular unit.

Thank you for your helpful comments. We added text to the introduction and discussion to address your point:

Introduction:

“The brain’s default mode network (DMN)—comprising the posterior medial cortex, medial prefrontal cortex, temporoparietal junction, and parts of anterior temporal cortex—was originally described as an intrinsic or “task-negative” network, activated when participants are not engaged with external stimuli (Raichle et al. 2001, Buckner et al 2008). […] Building on this foundation of prior work on the DMN, we asked whether we could find neural evidence for the retroactive influence of new knowledge on past memories.”

Discussion :

“In addition to mPFC, right precuneus and parts of temporal cortex exhibited significantly higher pattern similarity in the “twist” and “spoiled” groups who recalled the movie with the same interpretation. […] This transformation could affect how different regions and sub-systems of DMN represent memories, and suggests that the concerted activity of multiple subsystems and neural mechanisms might be at play during encoding, recall and successful updating of naturalistic event memories.”

2) I think that some direct comparison to regions outside the DMN would speak to whether the DMN is truly unique in carrying the key representations being discussed here. I was reluctant to suggest this because I think that the authors are justified in expecting that DMN regions would show the effects in question. However, there really is no "null" comparison here wherein a set of regions not expected to show these effects (e.g., a somatosensory network, or the frontoparietal network) in fact do not show them. There are not really controls or key differences being hypothesized across different conditions or regions. Rather, we have a set of regions that may or may not show pattern similarity differences to varying degrees, which feels very exploratory. The inclusion of some principled control comparisons, etc. would bolster these findings. The authors do include a whole-brain analysis in Supplementary Figure 1, which indeed produced many DMN regions. However, notably, regions outside the DMN such as the primary visual cortex and mid-cingulate cortex appear to show significant effects (which, based on the color bar, might actually be stronger than effects seen in the DMN). Given the specificity of the language in the paper in terms of the DMN, I think that some direct regional or network-level comparison is needed.

In the original submission, we included additional analyses for visual and somatosensory networks, which we hypothesized would serve as control networks. Following your comment, in the revision, we added a separate section (included below) more thoroughly examining these analyses. We also added text to the results and discussion to explain our interpretation of these findings.

Changes in neural representations beyond DMN

“We focused our core analyses on regions of the default mode network. Prior work has shown that multimodal neural representations of naturalistic events (e.g. movie scenes) are similar across encoding (movie-watching or story-listening) and verbal recall of the same events in the DMN (Chen et al., 2017; Zadbood et al., 2017). […] In the encoding-encoding comparison, several ROIs from the visual and somatomotor networks showed relatively strong effects as well (see Discussion).

In addition, we qualitatively reproduced our results by performing an ROI-based whole brain analysis (Appendix-Figure 3, p < 0.01 uncorrected). This analysis confirmed the importance of DMN regions for updating neural event representations. However, strong differences in pISC in the hypothesized direction were also observed in a handful of other non-DMN regions, including ROIs partly overlapping with anterior cingulate cortex and dorsolateral prefrontal cortex (see Discussion).”

Discussion:

“While our main goal in this paper was to examine how neural representations of naturalistic events change in the DMN, we also examined visual and somatosensory networks. […] These results suggest that both the "spoiled" manipulation and the "twist" may recruit top-down control and conflict monitoring processes during naturalistic viewing and recall.”

3) If I understand correctly, the main analyses of the fMRI data were limited to across-group comparisons of "critical scenes" that were maximally affected by the twist at the end of the movie. In other words, the analyses focused on the scenes whose interpretation hinged on the "doctor" versus "ghost" interpretation. I would be interested in seeing a comparison of "critical" scenes directly against scenes where the interpretation did not change with the twist. This "critical" versus "non-critical" contrast would be a strong confirmatory analysis that could further bolster the authors' claims, but on the other hand, it would be interesting to know whether the overall story interpretation led to any differences in neural patterns assigned to scenes that would not be expected to depend on differences in interpretation. (As a final note, such a comparison might provide additional analytical leverage for exploring the effect described in Figure 3B, which did not survive correction for multiple comparisons.)

This is a helpful suggestion, and we’ve added an analysis addressing your comment. We found that the interaction index capturing the difference between the three groups was stronger for the critical scenes than for the non-critical scenes for almost all DMN ROIs.

The role of scene content

“In the prior analyses, we focused on “critical scenes”, selected based on ratings from four raters who quantified the influence of the twist on the interpretation of each scene (see Methods). […] The interaction score across all DMN ROIs was significantly higher in “critical scenes” than all scenes (t(23) = 7.19, p = 2.53 x 10^-7^) and non-critical scenes (t(23) = 7.3, p = 1.95 x 10^-7^). These results show that critical scenes are indeed responsible for the observed pISC differences across groups.”

4) I appreciate the code being made available and that the neuroimaging data will be made available soon. I would also appreciate it if the authors made the movie stimulus and behavioral data available. The movie stimulus itself is of interest because it was edited down, and it would be nice for readers to be able to see which scenes were included.

Unfortunately due to copyright, we cannot share the movie stimulus outright. However, we will share the timing of the cuts used, as well as the time-stamped transcripts of verbal recall.

To sum up, I think that this is a great experiment with a lot of strengths. The design is fairly clean (especially for a movie stimulus), the analyses are well reasoned, and the data are clear. The only weaknesses I would suggest addressing are with regards to how the DMN is being described and evaluated, and the communication of how this work informs the field on a theoretical level.Reviewer #3 (Recommendations for the authors):I want to emphasize that I am a big fan of the study and the approach overall. It is very well done, the results are clear and interesting, and the paper is overall well constructed. Below, I will expand on some of the points I raised in the public review, and provide some suggestions for how they might be addressed.1) My first point dealt with asking for a somewhat stronger explanation for a focus on the DMN, and the question of how the reader should update their model of the DMN on the basis of these results. The updating phenomenon is certainly interesting, and as I noted in the public review, a focus on the DMN is sensible. However, I think this focus could be more clearly justified. The results themselves were a bit inconsistent in terms of which specific DMN regions showed pattern similarity effects, and I found myself wondering what this might mean. A bit more unpacking of this could be helpful. Beyond these points, however, there are theoretical frameworks arguing pretty compellingly that subsystems of the DMN may uniquely contribute to cognition (e.g., Maureen Ritchey's and Morris Moscovitch's ideas). This is a bit at odds with treating the DMN as a single unit. While I want to be clear that I do not think it is necessarily wrong to do so in this case, it does warrant some consideration.

Thank you for clarifying your points. We hope our added text in discussion addresses your concern.

2) As I noted in the public review, I hesitated to bring this up. However, some direct comparison of DMN vs. non-DMN would really bolster the results and the claims in my opinion. This would also go a long way in addressing my points above.

It was done.

3) I was not sure when writing this comment whether this "critical" versus "non-critical" analysis was tried, but in my view, it could serve to bolster the findings being presented in the paper currently. And as I noted in the public review, on the other hand, it would be interesting to know if the effects of the twist on neural patterns reached beyond the "critical" scenes into a general interpretation of the story.

This was a great suggestion and provided interesting results that we now include in the paper.